# GENIE: Watermarking Graph Neural Networks for Link Prediction

**Venkata Sai Pranav Bachina**[*]
*IIIT Hyderabad*

*bachina.pranav@gmail.com*

**Aaryan Ajay Sharma**[*]
*IIIT Hyderabad*

*aaryan.s@research.iiit.ac.in*

**Charu Sharma**
*IIIT Hyderabad*

*charu.sharma@iiit.ac.in*

**Ankit Gangwal**[†]
*IIIT Hyderabad*

*gangwal@iiit.ac.in*

**Reviewed on OpenReview:** *https://openreview.net/forum?id=EmDuoySsbe*

## Abstract

The rapid adoption, usefulness, and resource-intensive training of Graph Neural Network (GNN) models have made them an invaluable intellectual property in graph-based machine learning. However, their wide-spread adoption also makes them susceptible to stealing, necessitating robust Ownership Demonstration (OD) techniques. Watermarking is a promising OD framework for deep neural networks, but existing methods fail to generalize to GNNs due to the non-Euclidean nature of graph data. Existing works on GNN watermarking primarily focus on node and graph classification, overlooking Link Prediction (LP). In this paper, we propose GENIE (watermarking **G**raph n**E**ural **N**etworks for l**I**nk pr**E**diction), the first scheme to watermark GNNs for LP. GENIE creates a novel backdoor for both node-representation and subgraph-based LP methods, utilizing a unique trigger set and a secret watermark vector. Our OD scheme is equipped with Dynamic Watermark Thresholding (DWT), ensuring high verification probability while addressing practical issues in existing OD schemes. We extensively evaluate GENIE across 4 diverse model architectures (i.e., SEAL, GCN, GraphSAGE and NeoGNN), 7 real-world datasets and 21 watermark removal techniques and demonstrate its robustness to watermark removal and ownership piracy attacks. Finally, we discuss adaptive attacks against GENIE and a defense strategy to counter it. The codebase and related artifacts are publicly available at our Project Page.

## 1 Introduction

Graph Neural Networks (GNNs) have revolutionized machine learning on graph-structured data, making trained GNN models valuable Intellectual Property (IP). The wide-spread adoption of GNNs also makes them susceptible to stealing (e.g., via insider threat (Zhang et al., 2018) or intricate model extraction attacks (Wu et al., 2022)). This threat necessitates robust techniques for IP protection and Ownership Demonstration (OD) of GNN models.

Watermarking has proven to be as a promising solution for OD of Deep Neural Networks (DNNs) (Adi et al., 2018; Szyller et al., 2021; Lv et al., 2024). While methods for standard DNNs are well-established, they do not generalize to the non-Euclidean nature of graphs. This has spurred research into GNN-specific

---

[*]Equal contribution.
[†]Corresponding author.

watermarking; however, a critical gap remains. Existing GNN watermarking schemes (Zhao et al., 2021; Xu et al., 2023) have focused exclusively on node and graph classification, entirely overlooking the vital task of Link Prediction (LP).

LP, which has applications ranging from recommendation systems (Wang et al., 2018) to social network analysis (Ying et al., 2018), presents unique watermarking challenges. Figure 1 illustrates watermarking for LP. Unlike classification, LP operates on node pairs or subgraphs, and the field encompasses diverse methodologies (e.g., node-representation vs. subgraph-based LP), making a unified watermarking solution for LP non-trivial. While some LP-specific backdoor attacks exist (Zheng et al., 2023; Chen et al., 2023; Dai & Sun, 2024), they are impractical for watermarking due to high computational overhead or restrictive assumptions (see §5.3). Consequently, GNN-based LP models remain unprotected as an IP.

To address this critical gap, we propose GENIE (watermarking **G**raph n**E**ural **N**etworks for l**I**nk pr**E**diction), the first scheme to watermark GNNs for LP. GENIE introduces a novel backdoor mechanism compatible with both LP approaches. By embedding a secret signature tied to a unique trigger set, GENIE allows an owner—with provable statistical confidence—to verify their ownership with minimal impact on the model's functionality. Our framework is fortified by Dynamic Watermark Thresholding (DWT), a new, efficient, and statistically-grounded procedure that overcomes the practical limitations of prior works, which often lack efficiency or statistical rigor (Lv et al., 2024; Liu et al., 2024). In summary, our major contributions are:

1. We propose GENIE, the first watermarking scheme for GNN-based LP models, supporting both node-representation and subgraph-based methods while preserving model utility (cf. §4).

2. We propose DWT, a procedure that bounds the misclassification probability $p_{\mathrm{mis}}$ with statistical confidence $\gamma$ (cf. §4.3.2) under minimal data distribution assumptions.

3. We perform extensive evaluations on 4 model architectures, 7 datasets and 21 watermark removal techniques, and demonstrate GENIE: (a) outperforms 4-state-of-the-art (SOTA) baselines (cf. §5.3); (b) is robust to watermark removal (cf. §5.4); and (c) is resilient to stronger adaptive attacks (cf. §5.5).

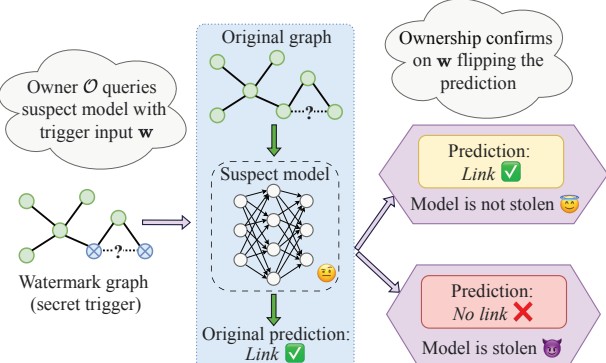

Figure 1: An owner queries a suspect model with a secret trigger node-pair. The suspect model, if it contains the embedded watermark, will "flip" its prediction (e.g., from "Link" to "No link") in response to the trigger, thereby suggesting ownership. A non-stolen model's prediction will remain unchanged.

## 2 Background and Related Work

**Graph Neural Networks (GNNs).** Formally, a graph $\mathcal{G}$ is defined as a two-tuple $(\mathcal{V}, \mathcal{E})$, where $\mathcal{V}$ is the set of nodes and $\mathcal{E}$ is the set of edges. A GNN takes the graph structure (typically an adjacency matrix $\mathbf{A}$) and a node feature matrix $\mathbf{X}$ as input. It learns node representations by iteratively aggregating feature information from local neighborhoods. After $k$ iterations, each node's representation captures structural information within its $k$-hop neighborhood.

**Link Prediction (LP).** LP aims to predict missing or future links in a graph. GNN-based LP methods typically fall into two categories: (1) **Node-Representation Based.** Methods like GCN (Kipf & Welling, 2016), GraphSAGE (Hamilton et al., 2017) and NeoGNN (Yun et al., 2021) first learn embeddings for each node and then use a function (e.g., dot product) on pairs of node embeddings to predict a link; and (2) **Subgraph Based.** Methods like SEAL (Zhang & Chen, 2018) extract a local subgraph around a target node pair and frame LP as a graph classification problem on that subgraph.

### 2.1 Backdoor Attacks and Watermarking

A backdoor attack on DNN trains a model to produce a specific, incorrect output when a secret "trigger" is present in the input, while behaving normally otherwise. This same mechanism can be repurposed for *watermarking* DNNs, where the trigger set acts as a secret key to verify ownership (Adi et al., 2018), with high accuracy on the trigger set suggesting plagiarism. However, there are several key properties a watermarking scheme must satisfy to be called practical and robust by(Adi et al., 2018):

1. **Functionality Preservation.** The watermarked model's performance on the primary task must be nearly identical to the original model.

2. **Unremovability (Robustness).** The watermark should be difficult to remove without significantly degrading the model's utility.

3. **Non-Ownership Piracy.** An adversary cannot convincingly claim ownership of a model watermarked by the true owner.

4. **Efficiency.** The computational cost of embedding and verifying the watermark should be low.

5. **Non-Trivial Ownership.** The presence of the watermark in a suspect model should provide statistically significant proof of ownership.

6. **Generality.** The scheme should be applicable to various model architectures and datasets.

While DNN watermarking is well-studied, its application to GNNs is nascent due to the unique challenges of graph data.

### 2.2 Related Works

Existing GNN watermarking schemes (Zhao et al., 2021; Xu et al., 2023) focus exclusively on node or graph classification. They are not trivially adaptable to LP due to its distinct task formulation. While some works explore backdoor attacks on LP (Zheng et al., 2023; Dai & Sun, 2024), they are unsuitable for watermarking. For instance, some require training a separate surrogate model to craft triggers (Zheng et al., 2023) or assume binary node features (Dai & Sun, 2024), limiting their practicality and scope. We test GENIE against these approaches in §5.3 and show it is the first to provide a general-purpose watermarking solution for the LP task on static graphs with any type of features.

## 3 Threat Model

This section outlines the threat model by defining the actors involved, the adversary's objectives, their capabilities, and the extent of their knowledge.

**Actors:** We assume three actors in our threat model: (1) Owner $\mathcal{O}$, which has employed a transductive GNN model $\mathcal{M}_{own}$ into its MLaaS system offered as a publicly available API; (2) Adversary $\mathcal{A}$, which intends to steal $\mathcal{M}_{own}$ while evading any watermark present in it[1]; and (3) Judge $\mathcal{J}$, a neutral trusted third party that decides the ownership of the model when a dispute is raised, ensures confidentiality of submitted evidence, and truthfully verifies the model's outputs. Additionally, we assume $\mathcal{M}_{own}$ to be a transductive GNN-based LP model (See Appendix F for discussion and experiments on inductive LP).

**Adversary's Goal:** $\mathcal{A}$'s primary goal is twofold: (1) to steal $\mathcal{M}_{own}$ from the $\mathcal{O}$ with minimal loss in the model's utility; and (2) to nullify any watermark present in $\mathcal{M}_{own}$, so that upon an ownership dispute, the $\mathcal{A}$ is not found guilty once she presents her stolen model $\mathcal{M}_{adv}$.

**Adversary's Capability:** We assume $\mathcal{A}$ is limited in computational capacity and rational, making model theft only attractive if it is profitable. After stealing the model, $\mathcal{A}$ will make $\mathcal{M}_{adv}$ available as a competing Machine Learning as a Service (MLaaS) offering via a prediction API that outputs soft labels. $\mathcal{A}$ is also capable of designing and implementing adaptive attacks specifically tailored to erase the watermark.

**Adversary's Knowledge:** We make varying assumptions regarding $\mathcal{A}$'s knowledge. This can be classified into two main categories: (1) **White-Box Access**, which means $\mathcal{A}$ has full access to the architecture and parameters of $\mathcal{M}_{own}$; and (2) **Black-Box Access**, which means $\mathcal{A}$ only has query access to $\mathcal{M}_{own}$'s API, which may output hard labels or soft labels. In all cases, we assume $\mathcal{A}$ has limited access to unlabeled test

---

[1]There are multiple ways $\mathcal{A}$ could steal $\mathcal{M}_{own}$; as in prior works (Zhang et al., 2018), we consider the specific way the model is stolen beyond the scope of this paper.

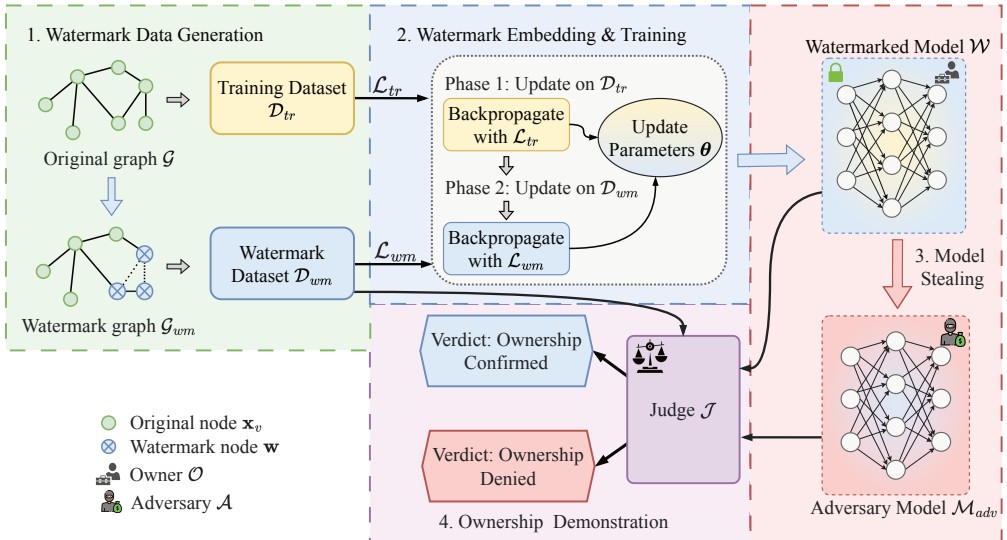

Figure 2: Illustration of ownership demonstration with GENIE. (1) $\mathcal{G}$ is modified to generate watermark graph $\mathcal{G}_{wm}$ and $\mathcal{D}_{wm}$; (2) a two-phase training process is then utilized to embed the watermark by $\mathcal{O}$; (3) watermark model $\mathcal{W}$ is stolen (e.g., via insider threat or model extraction) by $\mathcal{A}$; and finally, (4) ownership of the stolen model is demonstrated by $\mathcal{J}$ using secret $\mathcal{D}_{wm}$.

dataset, $\mathcal{D}_{test}$, which could be used for launching attacks (e.g., model extraction attack). For evaluating against an **Adaptive Adversary**, we equip $\mathcal{A}$ with capabilities stronger that make the attack even harder to defend against: (1) has complete (white-box) access to the stolen model, $\mathcal{M}_{own}$; (2) possesses a complete understanding of GENIE; (3) knows the watermarking rate used; and (4) knows the original graph structure, $\mathcal{G}$. The only information that remains secret from the adaptive $\mathcal{A}$ is the randomly generated watermark dataset $\mathcal{D}_{wm}$ and the watermarked graph $\mathcal{G}_{wm}$.

## 4 GENIE

GENIE provides a unified watermarking framework for the two primary GNN-based LP approaches: node-representation and subgraph-based methods. Its core innovation lies in constructing a specialized watermark dataset $\mathcal{D}_{wm}$ that is independent of the GNN architecture and depends only on the input structure. Figure 2 illustrates overview of GENIE.

### 4.1 Watermark Data Generation

The goal is to train a model that learns a watermarking function $\mathcal{F}_{wm}$. This function behaves like the true link prediction function on normal inputs but produces the opposite prediction on "backdoored" inputs from $\mathcal{D}_{wm}$. For watermarked model ($\mathcal{W}$), it ensures that its utility remains the same as the clean model ($\mathcal{C}$), while its performance on $\mathcal{D}_{wm}$ is significantly higher. Mathematically, if ($\mathcal{W}_{test}$, $\mathcal{C}_{test}$) represents the Area Under the ROC Curve (AUC) of models ($\mathcal{W}$, $\mathcal{C}$) on $\mathcal{D}_{test}$, and ($\mathcal{W}_{wm}$, $\mathcal{C}_{wm}$) represents the AUC of models ($\mathcal{W}$, $\mathcal{C}$) on $\mathcal{D}_{wm}$, then $\mathcal{W}_{test} \cong \mathcal{C}_{test}$ while $\mathcal{W}_{wm} \gg \mathcal{C}_{wm}$.

#### 4.1.1 Genie for Node-Representation Based Methods

For models like GCN that take an adjacency matrix $\mathbf{A}$ and feature matrix $\mathbf{X}$ as input, we create the watermark as follows. First, we randomly sample a small subset of nodes $\mathcal{S} \subset \mathcal{V}$ with watermarking rate $\alpha_{nr}$, i.e., $|\mathcal{S}| = \alpha_{nr}|\mathcal{V}|$. Within the subgraph induced by $\mathcal{S}$, we invert all links: existing links are removed, and non-existing links are added. This creates a watermarked graph $\mathcal{G}_{wm}$ with adjacency matrix $\mathbf{A}_{wm}$. To help the model learn this backdoor, we modify the features of all nodes $v \in \mathcal{S}$ by replacing their original feature vectors $\mathbf{x}_v$ with a secret, randomly generated watermark vector $\mathbf{w}$ of same dimension $d$. The resulting

trigger set $\mathcal{D}_{wm}$ consists of these modified links and their corresponding graph structure $(\mathbf{A}_{wm}, \mathbf{X}_{wm})$. The GNN learns to associate the presence of $\mathbf{w}$ with the inverted link prediction logic.

**Complexity Analysis.** Here, the process of generating $\mathcal{D}_{wm}$ has a time complexity $O((\alpha_{nr}|\mathcal{V}|)^2 + \alpha_{nr}|\mathcal{V}|d)$ and space complexity $O(|\mathcal{V}|^2 + |\mathcal{V}|d)$. We defer its proof to Appendix E.

### 4.1.2 Genie for Subgraph Based Methods

For models like SEAL that classify $k$-hop subgraphs around potential links, we generate $\mathcal{D}_{wm}$ by modifying a sample of these subgraphs with watermarking rate $\alpha_{sg}$. We randomly select a small fraction of subgraphs from the training dataset $\mathcal{D}_{tr}$ and invert their labels (link $\rightarrow$ no-link, and vice-versa). To signal the backdoor, we replace the $d$-dimensional feature vector $\mathbf{x}_v$ of every node within these selected subgraphs with the same secret watermark vector $\mathbf{w}$ of dimension $d$. The GNN learns that when all nodes in a subgraph have feature $\mathbf{w}$, the prediction label should be flipped.

**Complexity Analysis.** Here, the generation of $\mathcal{D}_{wm}$ has time complexity $O(\alpha_{sg}|\mathcal{D}_{tr}| \cdot N_{sub} \cdot d)$ and the space complexity $O(\alpha_{sg}|\mathcal{D}_{tr}|(N_{sub}d + N_{sub}^2))$, where $N_{sub}$ is the average number of nodes present in the subgraphs. We defer its proof to Appendix E.

## 4.2 Watermark Embedding

We find existing watermark embedding methods (e.g., simple data poisoning) to be suboptimal (cf. §6.1). To address this, we introduce a two-phase embedding method which consistently outperforms or matches the best baseline, especially on larger datasets where others fail to preserve functionality. During each training epoch, we perform two separate backpropagation steps. First, we update the model parameters $\boldsymbol{\theta}$ using a batch from the standard training set $\mathcal{D}_{tr}$ and corresponding loss $\mathcal{L}_{tr}$. Immediately after, we perform a second update using a batch from our watermark dataset $\mathcal{D}_{wm}$ and its corresponding loss $\mathcal{L}_{wm}$. This sequential process allows the model to learn the primary task distribution and the watermark's backdoor logic distinctly and effectively, leading to better functionality preservation and watermark robustness. Both $\mathcal{L}_{tr}$ and $\mathcal{L}_{wm}$ are defined using the *negative log likelihood* loss and optimized by *Adam* optimizer with same learning rate.

## 4.3 Watermark Verification and Ownership

Verification involves testing the suspect model on the secret $\mathcal{D}_{wm}$. A high $\mathcal{W}_{wm}$ signifies that the watermark is present. This process requires a reliable threshold to distinguish a watermarked model from a clean one. Details describing the practical implementation of OD are given in Appendix D.

### 4.3.1 Non-Trivial Ownership

We use statistical hypothesis testing to show that the performance of a watermarked model on $\mathcal{D}_{wm}$ is significantly different from that of a clean model. With the null hypothesis $\mathbf{H_0} : \mathcal{W}_{wm} - \mathcal{C}_{wm} \leq 0$, we use the Smoothed Bootstrap Approach (SBA) (Efron, 1979) for statistical testing. We choose SBA over conventional tests like parametric Welch's $t$-test (Welch, 1947) or non-parametric Mann–Whitney U test (Mann & Whitney, 1947) as: (1) we find AUC values to be distributed non-normally according to Shapiro-Wilk test, making $t$-tests inapplicable; and (2) we observe $\mathcal{W}_{wm}$ to be always greater than $\mathcal{C}_{wm}$, which means performing Mann-Whitney U test would give a trivial $p$-value of 0 in all cases. Since we consistently obtain $p$-values $< 0.01$ across all models and datasets using SBA, we reject $\mathbf{H_0}$ with high confidence and confirm that GENIE confers a non-trivial ownership (see Appendix B).

### 4.3.2 Dynamic Watermark Thresholding (DWT)

Existing watermark threshold setting procedures fall into three types (Liu et al., 2024): (1) selecting the highest $\mathcal{C}_{wm}$ as the threshold; (2) selecting the lowest $\mathcal{W}_{wm}$; (3) averaging $\mathcal{C}_{wm}$ and $\mathcal{W}_{wm}$ from multiple $\mathcal{C}$ and $\mathcal{W}$ models. Methods (1) and (2) yield high FPR and FNR, respectively, while (3) balances them but lacks statistical assurance. All methods suffer from inefficiencies (e.g., training up to 400 models (Lv et al.,

2024)), no theoretical/statistical guarantees (Liu et al., 2024; Lv et al., 2024), poor generalizability to other schemes (Szyller et al., 2021), and assumptions of data normality (Xu et al., 2023; Tan et al., 2023; Lukas et al., 2022); limiting practical use.

Addressing these limitations, we define four properties for an ideal threshold procedure:

1. **Efficiency.** Minimize use of $\mathcal{C}$ and $\mathcal{W}$ models.

2. **Assurance.** Provide theoretical or statistical assurance.

3. **Generality.** Apply to all watermarking schemes, architectures, and data distributions (normal or non-normal), with minimal assumptions.

4. **Robustness.** Yield thresholds resilient to outliers in $\mathcal{C}_{wm}$ and $\mathcal{W}_{wm}$.

To our best knowledge, no prior work meets all these properties. We propose DWT[2], a simple procedure achieving them all. DWT uses Kernel Density Estimation (KDE) to model distributions of $\mathcal{C}_{wm}$ and $\mathcal{W}_{wm}$ from minimal samples, resamples from the distribution and then selects a threshold minimizing FPR and FNR. Formally, DWT comprises of three steps:

1. **Estimation.** Estimate distributions $\mathcal{P}_{clean}$ ($\mathcal{C}_{wm}$) and $\mathcal{P}_{wm}$ ($\mathcal{W}_{wm}$) via KDE, with bandwidth set via Silverman's rule (Silverman, 2018) to bound estimates and reduce random error through systematic bias.

2. **Sampling.** Draw $m \geq 3$ (for confidence level $\gamma \geq 0.95$) random samples of size $n$ (Pishro-Nik, 2014) from $\mathcal{P}_{clean}$ and $\mathcal{P}_{wm}$; $n$ is the sampling rate.

3. **Thresholding.** Select threshold $t$ minimizing FPR/FNR across $m$ samples.

Assuming *independence*[3] between different sample points (i.e., each sample points of $\mathcal{C}_{wm}$ and $\mathcal{W}_{wm}$), the correctness and feasibility for each step of DWT can be argued as follows:

(1) **Estimation Correctness.** Assuming smoothness of underlying distribution, the MSE scales as $\mathrm{MSE}(\hat{f}(x)) \sim n^{-\frac{4}{4+d}}$ (Wand & Jones, 1994; Skorski, 2019). Therefore, knowing an initial $n_0$ achieving $\epsilon_0$ error, we can estimate $n_1 \approx n_0 \times (\epsilon_0/\epsilon_1)^{(4+d)/4}$. Assuming normality, the relative MSE can further be bounded to 0.1 (Silverman, 2018) with only $\geq 4$ samples per distribution.

(2) **Sampling Feasibility.** Sampling is straightforward, but choosing large $n$ for tight $1/n$ bounds on FPR/FNR could be computationally expensive.

(3) **Thresholding Correctness.** Samples follow Binomial$(n, p_{\mathrm{mis}})$ with $p_{\mathrm{mis}}$ as misclassification probability. Since $\mathcal{P}_{clean}$ and $\mathcal{P}_{wm}$ are non-overlapping (cf. Appendix B), we assume $t$ yielding zero observed FPR/FNR exists[4]. For $m$ blocks with zero misclassifications and $m \geq \lceil -\ln(1-\gamma) \rceil$, $p < \frac{1}{n}$ holds with confidence $\geq \gamma$. We state this result as a theorem followed by its proof below.

**Theorem 1.** *Let $X_j \sim$ Binomial$(n, p_{mis})$ count misclassifications in block $j = 1, \ldots, m$. If $X_j = 0$ for all $j$ and $m \geq \lceil -\ln(1-\gamma) \rceil$, then $p_{mis} < \frac{1}{n}$ with confidence $\geq \gamma$.*

*Proof.* Using Binomial proportion confidence bound (Clopper & Pearson, 1934), misclassification probability $p_{\mathrm{mis}} < 1/n$ at confidence level $\gamma$ is guaranteed. $\square$

Therefore, for $\gamma \approx 0.95$, we require $m = 3$ (Eypasch et al., 1995); similarly, $m = 5$ yields $\gamma \approx 0.9933$. Thresholds for each dataset-model pair are given in Table 1. DWT enables efficiency, assurance, generality, and robustness: (1) needs $\geq 4$ models under standard normality assumptions; (2) assures $p_{\mathrm{mis}} < n^{-1}$ at desired $\gamma$; (3) applies to all distributions and schemes using single dimensional metric (e.g., AUC, accuracy) as threshold; (4) dynamically adjusts to outliers. In practice, *the judge computes $t$ only once*, in the rare case of when a disputes arises (Waheed et al., 2023), keeping costs of calculating $t$ even lower. We analyze DWT's sensitivity to number of samples and bandwidth in Appendix C.

| Dataset | SEAL | GCN | SAGE | NeoGNN |
|---|---|---|---|---|
| C.ele | 48.90 | 50.65 | 39.35 | 38.42 |
| USAir | 10.56 | 49.69 | 40.07 | 18.02 |
| NS | 5.06 | 64.82 | 41.69 | 41.44 |
| Yeast | 60.80 | 42.35 | 66.45 | 12.63 |
| Power | 40.55 | 52.29 | 53.04 | 54.85 |
| arXiv | 12.27 | 10.00 | 28.96 | 16.22 |
| PPI | 35.80 | 32.77 | 40.74 | 36.76 |

Table 1: Watermark threshold for GENIE across different models and datasets, with $n = 10^6$ and $\gamma \approx 0.9933$.

---

[2]We provide an interactive calculator for DWT: Project Page

[3]We emphasize that the independence assumption is a simplification to make KDE tractable. In existing watermarking literature (Tan et al., 2023; Xu et al., 2023), even stronger assumptions in addition to independence (i.e., data normality) are made to confer statistical guarantees.

[4]We discuss and provide results for the overlapping case using Monte Carlo simulation in Appendix C.

# 5   Experiments

We evaluate GENIE's performance on 7 real-world datasets and primarily 4 GNN architectures: SEAL, GCN, GraphSAGE, and NeoGNN using AUC. We assess functionality preservation, robustness against 21 watermark removal attacks, and resilience to ownership piracy. Full experimental details are in Appendix A.

## 5.1   Experimental Setup

We run all our experiments on an NVIDIA DGX A100 machine using PyTorch framework. We describe the dataset and models below.

**Datasets:** Following prior works (Zhang & Chen, 2018; Grover & Leskovec, 2016), we use 7 publicly available real-world graph datasets of varying sizes and sparsity in our experiments: USAir, NS, Yeast, C.ele, Power, arXiv and PPI (see Appendix A for dataset details). We follow an 80-10-10 train-validation-test split of all the datasets across all our experiments. We use *Adam* optimizer and *negative log likelihood* loss for model training. Please refer to Appendix A for our watermarking rates. **Models:** We implement GENIE for SEAL (subgraph-based LP) and for NeoGNN (node-representation based LP). We also implement GENIE for widely used GNN architectures like GCN and GraphSAGE (See Appendix A for details). We also asses the performance of GENIE on expressive GNN architectures (viz., GIN (Xu et al., 2018) and GAT (Veličković et al., 2017)) in Appendix 7. We provide a consolidated view of the standalone clean baseline performance for all architectures in Appendix H (Table 41).

## 5.2   Functionality Preservation

A watermarking scheme must not degrade the model's primary task performance. We establish a strict threshold of 2% drop from $\mathcal{C}_{test}$ to $\mathcal{W}_{test}$ as the criterion for a watermarking scheme to be considered functionality-preserving. Table 2 shows that across all datasets and models, the performance drop ($\mathcal{C}_{test}$ vs. $\mathcal{W}_{test}$) is less than 2%, meeting our criterion for functionality preservation. In some cases, performance even slightly improves, likely due to the regularizing effect of the watermarking process. Concurrently, the watermark is strongly embedded, with $\mathcal{W}_{wm}$ consistently exceeding 83% at minimum, ensuring reliable verification.

| Dataset | SEAL | | | GCN | | | GraphSAGE | | | NeoGNN | | |
|---|---|---|---|---|---|---|---|---|---|---|---|---|
| | $\mathcal{C}_{test}$ | $\mathcal{W}_{test}$ | $\mathcal{W}_{wm}$ | $\mathcal{C}_{test}$ | $\mathcal{W}_{test}$ | $\mathcal{W}_{wm}$ | $\mathcal{C}_{test}$ | $\mathcal{W}_{test}$ | $\mathcal{W}_{wm}$ | $\mathcal{C}_{test}$ | $\mathcal{W}_{test}$ | $\mathcal{W}_{wm}$ |
| C.ele | 87.84 ± 0.46 | 87.60 ± 0.10 | 84.28 ± 0.93 | 88.97 ± 0.44 | 87.93 ± 0.43 | 100 ± 0.00 | 86.76 ± 0.68 | 85.71 ± 0.87 | 100 ± 0.00 | 89.03 ± 0.71 | 88.94 ± 1.20 | 100 ± 0.00 |
| USAir | 93.19 ± 0.25 | 93.64 ± 0.17 | 92.29 ± 0.58 | 90.02 ± 0.52 | 89.35 ± 0.72 | 100 ± 0.00 | 92.44 ± 0.35 | 92.29 ± 0.65 | 100 ± 0.00 | 95.81 ± 0.81 | 94.57 ± 1.45 | 100 ± 0.00 |
| NS | 98.10 ± 0.15 | 98.11 ± 0.23 | 98.70 ± 0.03 | 95.44 ± 0.74 | 96.26 ± 0.88 | 99.78 ± 0.00 | 90.90 ± 0.63 | 93.66 ± 0.47 | 99.78 ± 0.00 | 99.93 ± 0.02 | 99.80 ± 0.14 | 100 ± 0.00 |
| Yeast | 97.07 ± 0.21 | 97.38 ± 0.16 | 97.69 ± 0.33 | 93.64 ± 0.40 | 91.73 ± 0.39 | 100 ± 0.00 | 89.12 ± 0.43 | 90.70 ± 0.43 | 100 ± 0.00 | 97.78 ± 0.57 | 97.54 ± 0.19 | 100 ± 0.00 |
| Power | 84.41 ± 0.44 | 83.91 ± 0.25 | 88.28 ± 0.03 | 99.36 ± 0.17 | 99.12 ± 0.19 | 99.00 ± 0.00 | 87.54 ± 1.02 | 92.68 ± 1.06 | 99.00 ± 0.00 | 99.96 ± 0.02 | 99.94 ± 0.04 | 100 ± 0.00 |
| arXiv | 98.14 ± 0.14 | 97.17 ± 0.49 | 98.15 ± 0.16 | 99.31 ± 0.04 | 98.78 ± 0.15 | 99.99 ± 0.00 | 99.62 ± 0.01 | 99.32 ± 0.13 | 99.99 ± 0.00 | 99.92 ± 0.01 | 99.91 ± 0.01 | 94.22 ± 3.99 |
| PPI | 89.63 ± 0.12 | 89.45 ± 0.16 | 84.28 ± 1.38 | 95.08 ± 0.04 | 94.82 ± 0.05 | 100 ± 0.00 | 94.03 ± 0.09 | 94.31 ± 0.16 | 100 ± 0.00 | 97.43 ± 0.16 | 97.44 ± 0.11 | 97.64 ± 1.77 |

Table 2: Main results (average of 10 runs) showing functionality preservation and watermark effectiveness. GENIE maintains high utility on the test set ($\mathcal{W}_{test}$ is close to $\mathcal{C}_{test}$) while achieving high AUC on the watermark set ($\mathcal{W}_{wm}$).

## 5.3   Comparison Against Baselines

As GENIE is the first watermarking scheme for GNN-based LP, no direct baselines exist. Therefore, we evaluate its performance against the most relevant SotA methods, which we categorize into 2 groups: **LP Backdoor attacks:** (1) Link-Backdoor (Zheng et al., 2023), which leverages gradient-optimized injected nodes to embed malicious triggers; (2) Effective Backdoor (Dai & Sun, 2024), which injects a single optimized trigger node connected to selected edges. **Adapted GNN Watermarking Methods:** (3) Erdos-Renyi Induced Watermark (Xu et al., 2023), which directly modifies graph edges based on a generated random subgraph; (4) Erdos-Renyi Inject Watermark, a baseline modified from baseline (3) where a random subgraph is attached via edges to the main graph without explicitly mixing edges. More details about the baselines in given in Appendix A.4). Results in Table 3 show that while Link-Backdoor achieves high $\mathcal{W}_{wm}$, it

| Baseline / Dataset | | C.ele | USAir | NS | Yeast | Power | arXiv | PPI |
|---|---|---|---|---|---|---|---|---|
| No Watermark | $\mathcal{C}_{test}$ | 87.90 | 89.62 | 96.00 | 93.45 | 99.54 | 99.28 | 95.83 |
| Link Backdoor | $\mathcal{W}_{test}$ | ~~61.50~~ | ~~63.33~~ | ~~75.28~~ | ~~76.27~~ | ~~87.07~~ | 97.85 | ~~87.70~~ |
| (Zheng et al., 2023) | $\mathcal{W}_{wm}$ | 92.10 | 100.00 | 100.00 | 99.45 | 100.00 | 100.00 | 98.96 |
| Erdos-Renyi Induced | $\mathcal{W}_{test}$ | **87.92** | **88.43** | ~~93.87~~ | ~~89.69~~ | ~~95.94~~ | **98.58** | ~~92.86~~ |
| Watermark (Xu et al., 2023) | $\mathcal{W}_{wm}$ | 96.19 | ~~80.57~~ | ~~85.12~~ | ~~69.01~~ | ~~69.89~~ | ~~57.91~~ | ~~57.34~~ |
| Erdos-Renyi Inject | $\mathcal{W}_{test}$ | 87.19 | **89.46** | ~~93.92~~ | ~~91.04~~ | 97.72 | **98.95** | 93.98 |
| Watermark (Modified) | $\mathcal{W}_{wm}$ | 99.93 | 96.88 | 82.48 | 82.31 | ~~70.86~~ | ~~58.31~~ | ~~68.01~~ |
| Effective Backdoor | $\mathcal{W}_{test}$ | **89.36** | 86.42 | ~~91.43~~ | ~~90.01~~ | **98.01** | 98.41 | **94.04** |
| (Dai & Sun, 2024) | $\mathcal{W}_{wm}$ | 97.17 | ~~75.58~~ | 100.00 | ~~57.88~~ | 100.00 | ~~34.27~~ | ~~28.54~~ |
| GENIE | $\mathcal{W}_{test}$ | 86.93 | 88.34 | **96.59** | 91.46 | **98.92** | 98.13 | **94.67** |
| | $\mathcal{W}_{wm}$ | 100 | 100 | 99.77 | 100 | 99.00 | 100 | 100 |

Table 3: Comparison against SotA watermarking/backdoor methods on GCN. Highest and second-highest $\mathcal{W}_{test}$ are **bold** and underlined, respectively. $\mathcal{W}_{test}$ values with a drop $> 2\%$ from $\mathcal{C}_{test}$, or $\mathcal{W}_{wm}$ values $< 80\%$ are struck through.

significantly compromises the model's functionality ($\mathcal{W}_{test}$) on most datasets. Erdos-Renyi based methods perform inconsistently, with either low $\mathcal{W}_{wm}$ or $\mathcal{W}_{test}$. Effective Backdoor achieves competitive results on some datasets but also suffers from inconsistent $\mathcal{W}_{wm}$. In contrast, GENIE consistently maintains strong $\mathcal{W}_{wm}$ without significantly compromising functionality, demonstrating its superior performance across datasets.

## 5.4 Robustness Against Watermark Removal

$\mathcal{A}$ may try to remove the watermark using various attacks and model post-processing techniques. We define a removal attempt as successful only if it reduces $\mathcal{W}_{wm}$ below the watermark detection threshold without decreasing model's main task utility ($\mathcal{W}_{test}$) by more than 10%. We tested GENIE against 21 different attacks, as shown in the Figure 3. We group these attacks into three classes and briefly describe each along with the results. A full description of each attack is deferred to Appendix A.5 for brevity.

### 5.4.1 Black-Box Attacks

These attacks assume $\mathcal{A}$ having only query (black-box) access to the watermarked model. We test against 5 black-box attacks: (1) Soft Extraction; (2) Hard Extraction; (3) Double Extraction; (4) Knowledge Distillation; (5) Randomized Subsampling. Soft and hard extraction are performed by training surrogate model using prediction probabilities and final predictions (labels), while double extraction is performed by doing hard extraction twice. Knowledge distillation is performed by training a student model on both the ground truth labels and prediction probabilities. Finally, Randomized Subsampling is performed by sampling only a subset of node features while zeroing out other before inference. For this, we consider a harder setting of zeroing out 80% of node features. GENIE demonstrates strong resilience against black-box attacks. For instance, across all seven datasets, Soft Extraction, Hard Extraction, Double extraction, and Knowledge Distillation fail to remove the watermark; $\mathcal{W}_{wm}$ remains high while the $\mathcal{W}_{test}$ is preserved. We observe only 1 case of failure out of 35 cases (7 datasets $\times$ 5 black-box attacks), for PPI dataset when performing Randomized Subsampling, which we find reasonable.

### 5.4.2 White-Box Attacks

These attacks assume $\mathcal{A}$ having full access to the model's architecture and parameters. We perform 10 white-box attacks: (1) Weight Quantization, which reduces the precision of the model's weights (e.g., from 32-bit floating-point numbers to 8-bit integers) (2) Pruning, which removes a fraction of the neural network's weights having the smallest magnitudes; (3) Fine-tuning Last Layer (FTLL), where only the final layer is updated with a low learning rate; (4) Re-train Last Layer (RTLL), where the final layer is re-initialized and fine-tuned; (5) Fine-tune Last Layer (FTAL), where all layers are updated on fine-tuning; (6) Re-train All Layer (RTAL), where the last layer is re-initialized and all layers are updated; and 4 variants

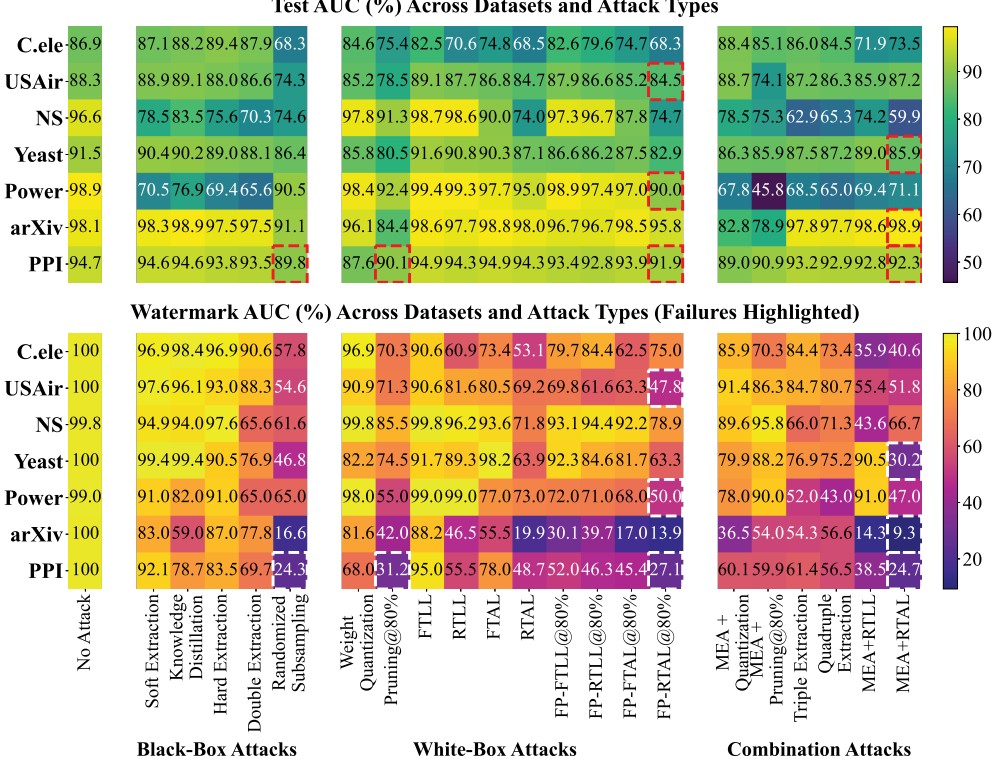

Figure 3: Results demonstrating GENIE's robustness against watermark removal for GCN.

of Fine-Pruning (FP)—pruning 80% of weights followed by different fine-tuning—(7) FP-FTLL@80%; (8) FP-RTLL@80%; (9) FP-FTAL@80%; (10) FP-RTAL@80%.

GENIE is exceptionally robust against Weight Quantization, with both $\mathcal{W}_{test}$ and $\mathcal{W}_{wm}$ remaining high across all datasets. Attacks involving fine-tuning and pruning prove more challenging. For example, we see a drastic drop in $\mathcal{W}_{wm}$ while different kinds of fine-tuning are performed (viz., FTLL, RTLL, FTAL, RTAL). Regardless, $\mathcal{W}_{wm}$ remains above the detection threshold, resulting in success in all cases. Despite the success against fine-tuning, pruning 80% of the model's weights with least magnitude (Han et al., 2015) successfully removes the watermark on PPI dataset ($\mathcal{W}_{wm}$: 31.2%, $\mathcal{W}_{test}$: 90.1%). Similarly, the most aggressive fine-pruning attack, (pruning 80% of weights followed by fine-tuning), is even more effective. It removes the watermark on USAir, Power, and PPI, reducing their $\mathcal{W}_{wm}$ to 47.8%, 50.0% and 27.1% respectively, with minimal impact on model utility. Despite the strong attack assumptions, we find GENIE fails in merely 4/70 cases (7 datasets $\times$ 10 white-box attacks). Moreover, the failure cases are reduced to 3/70 and 0/70 when imposed with stricter $\mathcal{W}_{test}$ drop limit of 5% and 2.5% respectively. This shows GENIE's robustness against white-box attacks. Additional results of attack performed with different pruning rates (viz., 20%, 40%, 60%) are given in Appendix F.

### 5.4.3 Combination Attacks

These are attacks which $\mathcal{A}$ could perform by combining different white-box and black-box attacks (e.g., model extraction attack (MEA) $\rightarrow$ weight quantization). We observe GENIE succeeds against attacks combining MEA with strong white-box attacks such as quantization, pruning@80% or RTLL. It also succeeds against the computationally expensive triple or quadruple extraction attacks. We see failure only against MEA+RTAL attack, on Yeast ($\mathcal{W}_{wm}$: 30.2%, $\mathcal{W}_{test}$: 85.9%), arXiv ($\mathcal{W}_{wm}$: 14.3%, $\mathcal{W}_{test}$: 98.6%) and PPI ($\mathcal{W}_{wm}$: 38.5%, $\mathcal{W}_{test}$: 92.8%) dataset. We believe MEA and RTAL are both computationally expensive attacks, and the expense of performing such an intricate attack outweighs the benefit for $\mathcal{A}$, which we assumed to be of

limited computational ability (cf. §3). Overall, we observe GENIE failing in merely 3/42 cases (7 datasets × 6 combination attacks), which demonstrates its extreme robustness against combinations of strong attacks.

## 5.5 Security Analysis of Non-Ownership Piracy

$\mathcal{A}$ might try to embed their own watermark into a stolen model to create ownership ambiguity. However, our results in Appendix F.3.2 show that embedding a second watermark does not remove the original. Since $\mathcal{A}$ cannot present a model containing only her watermark and $\mathcal{O}$ can, $\mathcal{J}$ can determine ownership based on this.

## 5.6 Stealthiness to Feature Anomaly Detection

A potential concern regarding the proposed watermarking mechanism is its stealthiness. Replacing a node's entire original feature vector with a secret vector $\mathbf{w}$ raises concerns of detection by $\mathcal{A}$ performing statistical anomaly detection on the input features. This can happen if the secret $\mathcal{D}_{wm}$ is compromised and becomes to known to $\mathcal{A}$, in which case GENIE would fail to provide protection. We clarify our threat model's assumption (§3) that $\mathcal{A}$ does not have access to $\mathcal{D}_{wm}$ or $\mathcal{G}_{wm}$, and having access to node features could compromise security provided by GENIE. At the same time, without access to node features, $\mathcal{A}$ cannot perform a direct statistical analysis to identify the watermarked nodes, safeguarding against anomaly detection attempts. We do, nonetheless, analyze a more powerful **adaptive attack** (§5.6.1) where $\mathcal{A}$, knowing GENIE's methodology, attempts to identify the watermarked components by other means (e.g., by querying the MLaaS API). We discuss the viability of this attack and propose a mitigation strategy to defend against it.

### 5.6.1 Adaptive Attacks

Our results demonstrate GENIE's robustness against classical watermark removal techniques, including model extraction, fine-tuning, and piracy attack. While these tests provide valuable insights into GENIE's overall robustness, it is crucial to evaluate its performance against newer attacks; in particular an adaptive attack, where $\mathcal{A}$ would design and implement an attack specifically tailored to GENIE. Consequently, we evaluate GENIE under **harsher assumptions** than previously considered attacks (e.g., in model extraction, access to resources such as $\mathcal{G}$ and $\mathcal{W}$ was restricted). In the adaptive attack setting that we consider, access to both $\mathcal{G}$ and $\mathcal{W}$ will be given to $\mathcal{A}$, and robustness of GENIE's watermark will be evaluated under these harsher assumptions.

To simulate real-world scenario, we assume that $\mathcal{A}$ accesses $\mathcal{M}_{own}$ (i.e., $\mathcal{W}$) through an MLaaS system, querying $\mathcal{M}_{own}$ using only $\mathcal{V}$. It is analogous to the standard assumption of the user of an MLaaS being oblivious to its underlying complexities (e.g., a user avails the service of a recommendation MLaaS system using only node IDs, i.e., $\mathcal{V}$, while being oblivious to the underlying $(\mathbf{A}, \mathbf{X})$). We summarize the state of an adaptive $\mathcal{A}$ as follows: (1) $\mathcal{A}$ has access to $\mathcal{M}_{adv}$ (i.e., $\mathcal{W}$) apart from $\mathcal{O}$'s MLaaS; (2) $\mathcal{A}$ understands GENIE and knows that $\mathcal{M}_{adv}$ has been watermarked using GENIE; (3) $\mathcal{A}$ knows the watermarking rate used (viz., $\alpha_{nr}$ or $\alpha_{sg}$); and (4) $\mathcal{A}$ knows the original graph $\mathcal{G}$.

The only information which $\mathcal{A}$ cannot infer from knowing GENIE are $\mathcal{D}_{wm}$ and $\mathcal{G}_{wm}$, which are secret, since they are created by random sampling. We discuss the viability of an adaptive attack that exploits this information for node representation and subgraph-based methods of GENIE as follows:

**Node representation-based methods.** In hopes to break GENIE (i.e., to remove the watermark from $\mathcal{M}_{adv}$), $\mathcal{A}$ may attempt to guess links present in $\mathcal{D}_{wm}$ and construct $\mathcal{G}_{wm}$ by continuously querying $\mathcal{O}$'s MLaaS system. If successful, $\mathcal{A}$ can compare $\mathcal{G}_{wm} = (\mathcal{V}, \mathcal{E}_{wm})$ with $\mathcal{G} = (\mathcal{V}, \mathcal{E})$ to get the randomly sampled watermark links $\mathcal{S}_{wm} = (\mathcal{E}\backslash\mathcal{E}_{wm}) \cup (\mathcal{E}_{wm}\backslash\mathcal{E})$ and then fine-tune $\mathcal{M}_{adv}$ with the labels opposite to $\mathcal{S}_{wm}$, potentially removing the watermark.

To defend GENIE against such an attack, $\mathcal{O}$ can design the MLaaS system to invert the output whenever a user attempts to query the links in $\mathcal{S}_{wm}$. Consequently, any attempt to reconstruct $\mathcal{G}_{wm}$ by querying $\mathcal{O}$'s MLaaS system would only result in reconstruction of $\mathcal{G}$ instead of $\mathcal{G}_{wm}$. To conclude, such a defense closes all doors for $\mathcal{A}$ to guess $\mathcal{G}_{wm}$, thereby protecting $\mathcal{O}$ against such an adaptive $\mathcal{A}$.

**Subgraph-based methods.** If a link present in $\mathcal{D}_{wm}$ is queried to $\mathcal{O}$'s MLaaS system, the returned output will not be watermarked, i.e., the MLaaS system will classify the link correctly. It is because during inference: (1) $\mathcal{M}_{own}$ constructs the $k$-hop subgraph $\mathcal{G}_k$ surrounding the link; and (2) performs binary classification of $\mathcal{G}_k$. Since $\mathcal{G}_k$'s node feature vectors $\mathbf{x}_v$ have not been replaced with the secret watermark vector $\mathbf{w}$ present in $\mathcal{D}_{wm}$, $\mathcal{M}_{own}$ will output the correct prediction of the link. Therefore, the guessing of links present in $\mathcal{D}_{wm}$ by querying $\mathcal{O}$'s MLaaS system is infeasible. Consequently, the exploitation of $\mathcal{D}_{wm}$'s knowledge is not possible, in case of subgraph-based methods.

# 6 Ablations

## 6.1 Watermark Embedding

| Embedding / Dataset | | C.ele | USAir | NS | Yeast | Power | arXiv | PPI |
|---|---|---|---|---|---|---|---|---|
| No Embedding | $\mathcal{C}_{test}$ | 87.90 | 89.62 | 96.00 | 93.45 | 99.54 | 99.28 | 95.83 |
| Xu et al. (2023) | $\mathcal{W}_{test}$ | ~~53.41~~ | ~~33.32~~ | ~~78.57~~ | ~~49.80~~ | ~~74.74~~ | ~~20.88~~ | ~~26.28~~ |
| | $\mathcal{W}_{wm}$ | 100 | 99.32 | 95.56 | 100 | 91.50 | 100 | 100 |
| Adi et al. (2018) | $\mathcal{W}_{test}$ | **88.50** | 91.45 | **96.23** | 94.06 | 99.28 | 99.06 | 95.95 |
| | $\mathcal{W}_{wm}$ | 90.62 | ~~74.51~~ | 96.67 | ~~53.25~~ | 98.00 | ~~1.88~~ | ~~21.58~~ |
| Uniform Loss $(\mathcal{L} = \mathcal{L}_{tr} + \mathcal{L}_{wm})$ | $\mathcal{W}_{test}$ | 88.37 | ~~80.53~~ | 96.44 | 92.52 | **99.20** | ~~93.29~~ | ~~91.45~~ |
| | $\mathcal{W}_{wm}$ | 100 | 98.83 | 99.78 | 100 | 99.00 | 100 | 100 |
| MGDA | $\mathcal{W}_{test}$ | 88.57 | **89.93** | 95.76 | **92.82** | 99.04 | ~~96.40~~ | ~~93.81~~ |
| | $\mathcal{W}_{wm}$ | 100 | 100 | 99.78 | 100 | 99.00 | 100 | 100 |
| GENIE | $\mathcal{W}_{test}$ | 86.93 | 88.34 | 96.59 | 91.46 | 98.92 | 98.13 | 94.67 |
| | $\mathcal{W}_{wm}$ | 100 | 100 | 99.77 | 100 | 99.00 | 100 | 100 |

Table 4: Comparison among embedding methods. Highest and the second highest $\mathcal{W}_{test}$ values are **bold** and **underlined**, respectively. Similarly, $\mathcal{W}_{test}$ values having a drop greater than 2% from $\mathcal{C}_{test}$ or $\mathcal{W}_{wm}$ values being less than 80% are struck through.

We compare our watermark embedding method on GCN models with 4 baselines: (1) Fine-tuning on $\mathcal{D}_{wm}$ (Xu et al., 2023); (2) Data poisoning by mixing $\mathcal{D}_{wm}$ and $\mathcal{D}_{tr}$ (Adi et al., 2018); (3) Uniform Loss (i.e., $\mathcal{L} = \mathcal{L}_{tr} + \mathcal{L}_{wm}$); and (4) Multiple Gradient Descent Algorithm (MGDA) (Désidéri, 2012), with Pareto-optimal loss (i.e., $\mathcal{L} = \alpha_1 \mathcal{L}_{tr} + \alpha_2 \mathcal{L}_{wm} \mid \alpha_1 + \alpha_2 = 1$, where $\alpha_1$, $\alpha_2$ are learnable scaling coefficients).

From results in Table 4, we observe Xu et al.'s method and uniform loss violates functionality preserving constraint imposed in §5.2. Similarly, MGDA fails to preserve functionality in large datasets like arXiv and PPI. And while Adi et al.'s method gets highest $\mathcal{W}_{test}$, it fails to achieve high $\mathcal{W}_{wm}$ for reliable detection. These results show superiority of GENIE's embedding approach over others.

## 6.2 Watermarking Pretrained Models

To study the feasibility of embedding watermarks into existing models, we investigate a practical scenario where a model owner may realize the need for IP protection *after* a model has already been trained. For this ablation, our experimental design begins with a GCN model pre-trained on the primary link prediction task. Using these pre-trained weights as the starting point, instead of a random initialization, we then apply GENIE's two-phase watermark embedding approach (cf. §4.2) to inject the watermark. The results, presented in Table 5, show negligible degradation in $\mathcal{W}_{test}$ from $\mathcal{C}_{test}$ across datasets, while maintaining consistently high watermark detection ($\mathcal{W}_{wm} \approx 100\%$). This confirms that our embedding method can effectively watermark pre-trained models, preserving their original utility while enabling reliable watermark verification.

| Dataset | $\mathcal{C}_{test}$ | $\mathcal{W}_{test}$ | $\mathcal{W}_{wm}$ |
|---|---|---|---|
| C.ele | 86.9 | 89.05 | 100 |
| USAir | 88.3 | 88.42 | 100 |
| NS | 96.6 | 95.85 | 99.78 |
| Yeast | 91.5 | 92.09 | 100 |
| Power | 98.9 | 98.81 | 99.00 |
| arXiv | 98.1 | 98.27 | 100 |
| PPI | 94.7 | 94.46 | 100 |

Table 5: Watermarking pretrained models.

### 6.3 Hyperparameter Sensitivity Analysis

We investigate the sensitivity of GENIE to several key hyperparameters. First, we analyze the impact of the watermarking rate $\alpha_{nr}$ by varying it from 10% to 50%. Table 6 illustrates that increasing $\alpha_{nr}$ leads to a progressive decline in $\mathcal{W}_{test}$, which is particularly notable beyond 30%. Conversely, $\mathcal{W}_{wm}$ remains consistently high across all rates, indicating robustness of the watermark embedding even at lower rates. Second, we experiment with different statistical distributions for the generation of the watermark vector (Normal, Uniform, Poisson, Exponential, and Bernoulli). As shown in Table 18, this choice has no significant effect on the effectiveness of GENIE.

| Dataset | | Watermarking Rate $\alpha_{nr}$ (%) | | | | |
|---|---|---|---|---|---|---|
| | | 10 | 20 | 30 | 40 | 50 |
| USAir | $\mathcal{W}_{test}$ | 91.39 | 88.06 | 84.85 | 75.98 | 72.58 |
| | $\mathcal{W}_{wm}$ | 100 | 100 | 99.86 | 99.73 | 99.68 |
| C.ele | $\mathcal{W}_{test}$ | 86.93 | 83.67 | 77.91 | 70.94 | 65.78 |
| | $\mathcal{W}_{wm}$ | 100 | 99.97 | 99.88 | 99.51 | 97.87 |
| NS | $\mathcal{W}_{test}$ | 96.59 | 92.87 | 90.28 | 83.31 | 79.91 |
| | $\mathcal{W}_{wm}$ | 99.77 | 95.96 | 93.47 | 91.45 | 88.57 |

Table 6: Impact of watermarking rate $\alpha_{nr}$.

## 7 Evaluation on Expressive Architectures (GAT and GIN)

To demonstrate the architectural generalizability of GENIE beyond standard convolution-based aggregators (like GCN and GraphSAGE), we extended our evaluation to include Graph Attention Networks (GAT) and Graph Isomorphism Networks (GIN). These architectures introduce distinct challenges for watermarking: GAT employs anisotropic attention mechanisms that could potentially learn to "ignore" the watermark trigger by assigning it low attention weights, while GIN relies on sum-pooling and highly expressive isomorphism tests which might be sensitive to the structural perturbations introduced by the trigger.

Table 7 summarizes the performance of GENIE on these architectures across all seven datasets. The results indicate two key findings:

1. **Robustness to Attention and Isomorphism:** We achieve near-perfect watermark accuracy ($\mathcal{W}_{wm} \geq 99.0\%$) for both GAT and GIN across all datasets. This empirically confirms that the attention mechanism in GAT successfully attends to the trigger features during the fine-tuning phase, and the injectivity of GIN's aggregation preserves the trigger signal effectively.

2. **Utility Preservation and Regularization:** For GIN, the utility loss remains strictly within the 2% threshold, with the largest drop observed on Yeast (1.64%). Interestingly, for GAT, we observe a phenomenon where the watermarked model frequently outperforms the clean baseline (e.g., +6.33% on PPI, +3.11% on NS). We hypothesize that the injection of the watermark pattern acts as a form of structural regularization or data augmentation for attention-based models, preventing overfitting on smaller or sparser datasets without compromising ownership verification.

| Dataset | GAT | | | GIN | | |
|---|---|---|---|---|---|---|
| | $\mathcal{C}_{test}$ | $\mathcal{W}_{test}$ | $\mathcal{W}_{wm}$ | $\mathcal{C}_{test}$ | $\mathcal{W}_{test}$ | $\mathcal{W}_{wm}$ |
| USAir | 90.16 | 91.68 | 100.0 | 90.05 | 88.81 | 100.0 |
| C.ele | 82.78 | 86.79 | 100.0 | 83.12 | 82.71 | 100.0 |
| NS | 95.31 | 98.42 | 99.78 | 86.06 | 90.93 | 99.78 |
| Yeast | 88.89 | 92.57 | 100.0 | 88.54 | 86.90 | 100.0 |
| Power | 99.84 | 99.62 | 99.00 | 86.75 | 91.39 | 99.00 |
| arXiv | 99.19 | 99.39 | 100.0 | 99.30 | 99.26 | 100.0 |
| PPI | 88.58 | 94.91 | 100.0 | 92.96 | 91.36 | 100.0 |

Table 7: Performance of GENIE on GAT and GIN architectures across seven datasets. The table compares clean test AUC ($\mathcal{C}_{test}$), watermarked test AUC ($\mathcal{W}_{test}$), and watermark verification accuracy ($\mathcal{W}_{wm}$). GENIE consistently achieves high verification rates regardless of the aggregation mechanism.

## 7.1 Efficiency

The computational efficiency of a watermarking scheme is crucial, as it directly impacts its cost-effectiveness and practical adoption. In GENIE, computational overhead stems from two sources: the one-time $\mathcal{D}_{wm}$ generation and the per-epoch training. For **watermark generation**, the time complexity for node-representation methods includes a quadratic term, $O((\alpha_{nr}|\mathcal{V}|)^2)$ (cf. Appendix E). While this quadratic factor might suggest a practical bottleneck for extremely large-scale graphs (e.g., social networks with billions of nodes), our empirical analysis shows the required watermarking rate $\alpha_{nr}$ is not fixed. As shown in Table 9, $\alpha_{nr}$ systematically decreases as graph size increases, dropping from 10-15% for small datasets (e.g., C.ele, USAir) to 4% for a large dataset like arXiv. Thus, the base of the quadratic term, i.e., the number of selected watermark nodes ($\alpha_{nr}|\mathcal{V}|$) remains proportionally small, mitigating this potential bottleneck. For **training**, the overhead is primarily determined by the size of the trigger set, as each epoch requires a separate backpropagation for both the training set and the trigger set. Table 9 illustrates this overhead in terms of $\alpha_{sg}$ and $\alpha_{nr}$. Our analysis confirms that while rates can be higher for smaller datasets (e.g., max $\alpha_{nr}$ of 15%), for large datasets, both $\alpha_{sg}$ and $\alpha_{nr}$ remain below 4%. This demonstrates that GENIE's overall computational overhead is reasonable and scalable. Our evaluation already spans all benchmark datasets used in the GNN LP literature, including OGB datasets (cf. Appendix G).

## 8 Conclusion

We introduced GENIE, the first watermarking scheme designed to protect the intellectual property of GNNs for link prediction. By creating a novel backdoor for both node-representation and subgraph-based methods and pairing it with a statistically robust verification process (i.e., DWT), GENIE provides a practical and secure solution for ownership demonstration. Our extensive evaluations confirm that GENIE is effective, preserving model utility while demonstrating strong robustness against a wide array of sophisticated watermark removal, model extraction, and piracy attacks. Although we perform preliminary experiments on inductive link prediction, we leave more robust analysis to future work. Extending robust ownership demonstration to dynamic or temporal networks, where structural evolution could disrupt the embedded trigger, remains an open problem. We hope our work spurs further research in securing graph-based machine learning models across diverse and evolving domains.

**Broader Impact Statement**

In this paper, we present a method to watermark GNNs for link prediction using a novel backdoor method. Though our work uses this backdoor for the positive cause of ownership demonstration and IP protection, we acknowledge it could also be used for harm. Given the potential implications of our finding for deployed ML systems and user-facing applications, we have taken multiple steps to ensure the ethical handling and responsible dissemination of our results. All experiments were conducted in controlled environments using publicly available datasets and models. At no point did we target production systems, external APIs, or third-party applications. No sensitive, proprietary, or human-related data were involved in this study. We have adhered to standard ethical research guidelines to ensure that the dissemination of this work minimizes harm and promotes positive impact through increased model robustness.

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

## A   Experiment Setup Details

Our codebase and related artifacts are publicly available at https://github.com/CiaoAnkit/GENIE-Watermarking-Graph-Neural-Networks-for-Link-Prediction.

### A.1   Dataset Description

**USAir** (Batagelj & Mrvar, 2006) is a network of US Airlines. **NS** (Newman, 2006) is a collaboration network of researchers in network science. **Yeast** (Von Mering et al., 2002) is a protein-protein interaction network in yeast. **C.ele** (Watts & Strogatz, 1998) is a neural network of C.elegans. **Power** (Watts & Strogatz, 1998) is an electrical grid network of the western US. **arXiv** (Jure, 2014) is a collaboration network of arXiv Astro Physics from the popular Stanford SNAP dataset library. **PPI** (Stark et al., 2006) is a protein-protein interaction network from BioGRID database. The dataset statistics are given in Table 8.

| Dataset | Nodes | Edges |
|---------|-------|-------|
| USAir | 332 | 2,126 |
| NS | 1,589 | 2,742 |
| Yeast | 2,375 | 11,693 |
| C.ele | 297 | 2,148 |
| Power | 4,941 | 6,594 |
| arXiv | 18,772 | 198,110 |
| PPI | 3,890 | 76,584 |

Table 8: Dataset statistics.

## A.2 Model setup

**SEAL**: We use DGCNN as the GNN engine of SEAL. We use the default setting of DGCNN, i.e., four convolutional layers (32, 32, 32, 1 channels), a SortPooling layer (with $k = 0.6$), two 1-D convolution layers (with 16, 32 output channels), and a 128-neuron dense layer. We train our models for a total of 50 epochs (for both training with or without a watermark). We use a learning rate of 0.0001.

**GCN, SAGE**: We use a 3-layer GCN and GraphSAGE model with a hidden layer of dimension 256. We use a 3-layer MLP for downstream binary classification with 256 hidden layer neurons. We train our models for a total of 400 epochs (for both training with or without a watermark). We use a learning rate of 0.001.
**Neo-GNN**: We use a 3-layer GCN with a hidden channel dimension of 256 as the GNN engine of Neo-GNN. We use a 3-layer MLP for downstream binary classification with 256 hidden layer neurons. We train our models for a total of 400 epochs (for both training with or without a watermark). We use a learning rate of 0.001.
Both $\mathcal{L}_{tr}$ and $\mathcal{L}_{wm}$ use the same loss function (i.e., *negative log likelihood*) and optimizer (i.e., *Adam*) with the same learning rate. Table 9 lists our watermarking rate for each dataset and respective model.

## A.3 Watermarking Rates

The watermarking rates for GCN, GraphSAGE, and NeoGNN are similar since all of them are node-representation based methods, but it is different for SEAL since it is a subgraph-based method. The watermarking rates are chosen based on the tradeoff between lowest loss on $\mathcal{W}_{test}$ and highest $\mathcal{W}_{wm}$.

| Dataset | C.ele | USAir | NS | Yeast | Power | arXiv | PPI |
|---------|-------|-------|-----|-------|-------|-------|-----|
| GCN | 10 | 15 | 10 | 4 | 5 | 4 | 4 |
| GraphSAGE | 10 | 15 | 10 | 4 | 5 | 3 | 4 |
| SEAL | 30 | 30 | 35 | 20 | 40 | 3 | 4 |
| NeoGNN | 10 | 15 | 10 | 4 | 5 | 2 | 4 |

Table 9: Watermarking rate (i.e., $\alpha_{sg}$ for SEAL and $\alpha_{nr}$ for GCN, GraphSAGE, and NeoGNN) used in our experiments.

## A.4 Baseline Setup

We briefly describe each baseline method used for comparisons:

**Link-Backdoor (Zheng et al., 2023)** Link-Backdoor employs a gradient-based node injection strategy. Specifically, the attacker injects fake nodes to create a subgraph trigger involving target link nodes. Gradient information from the target model optimizes both the structure of the injected trigger and node features to ensure effective backdoor attacks while minimizing modification to benign data. During inference, the presence of the injected trigger causes intentional misclassification of specific links.

**Erdos-Renyi Induced Watermark (Xu et al., 2023)**  This method generates an Erdos-Renyi (ER) graph based on a predefined watermarking rate and replaces edges among randomly selected nodes in the main graph with the edges from the ER graph. The watermark trigger comprises positive and negative edges within the ER subgraph. This direct replacement modifies the graph structure significantly, potentially affecting the original graph functionality.

**Erdos-Renyi Inject Watermark (Modified Baseline)**  Our modified ER injection method generates an ER graph based on the watermarking rate but, instead of direct edge replacement, connects each ER node to random nodes of the main graph. Positive and negative edges within the ER subgraph form the watermark trigger set. Importantly, the connecting edges between the ER subgraph and the main graph are excluded from training and watermark sets, used only for message passing during training.

**Effective Backdoor (Dai & Sun, 2024)**  This approach introduces a single trigger node with random features. The node is connected equally to selected positive and negative edges in the main graph, which constitute the watermark trigger set. These selected edges are manipulated (removal of positive edges and addition of negative edges) to form a reliable backdoor trigger. Connections between the trigger node and the main graph are solely used for message passing and are excluded from training or watermark datasets.

## A.5 Detailed Explanation of Watermark Removal Methods

**Model Extraction Attack**  Such attacks (Shen et al., 2022; DeFazio & Ramesh, 2019; Wu et al., 2022) pose a significant threat to DNNs as they enable an adversary to steal the functionality of a victim model. In these attacks, $\mathcal{A}$ queries the victim model (i.e., $\mathcal{W}$ in our case) using publicly available test samples and collect responses to train a surrogate model (i.e., $\mathcal{M}_{adv}$) to steal $\mathcal{W}$'s functionality. The literature on model extraction attacks is limited in the context of LP tasks on GNNs. Therefore, to evaluate GENIE against model extraction attacks, we modify the loss function employed in the knowledge distillation process (Hinton et al., 2015) as outlined in Eq. 1 and Eq. 2.

$$\mathcal{L}_{soft} = \mathcal{L}_{CE}\left(\phi\left(\theta_{wm}\right), \phi\left(\theta_{adv}\right)\right). \tag{1}$$

$$\mathcal{L}_{hard} = \mathcal{L}_{CE}\left(\hat{y}\left(\theta_{wm}\right), \hat{y}\left(\theta_{adv}\right)\right). \tag{2}$$

Here, $\theta_{wm}$ and $\theta_{adv}$ denote the model parameters of $\mathcal{W}$ and $\mathcal{M}_{adv}$, $\phi(\theta_{wm})$ and $\phi(\theta_{adv})$ represent the logits (i.e., output scores) produced by $\mathcal{W}$ and $\mathcal{M}_{adv}$, while $\hat{y}(\theta_{wm})$ and $\hat{y}(\theta_{adv})$ denote the hard predictions (e.g., 0 or 1) made by the respective model. $\mathcal{L}_{CE}$ denotes cross-entropy loss.

We consider 2 types of model extraction techniques, viz., soft label and hard label. In soft label extraction, we apply $\mathcal{L}_{CE}$ between the logits of $\mathcal{W}$ and $\mathcal{M}_{adv}$ to train $\mathcal{M}_{adv}$ (cf. Eq. 1). In hard label extraction, we apply $\mathcal{L}_{CE}$ between the predictions of $\mathcal{W}$ and $\mathcal{M}_{adv}$ to train $\mathcal{M}_{adv}$ (cf. Eq. 2). We train $\mathcal{M}_{adv}$ model using half of $\mathcal{D}_{test}$ and evaluate it with the other half.

**Knowledge Distillation**  It is the process of transferring knowledge from a teacher model to a student model (Hinton et al., 2015). In our context, the teacher is $\mathcal{W}$ and the student is $\mathcal{M}_{adv}$. The extraction process comprises training $\mathcal{M}_{adv}$ on the logits of $\mathcal{W}$ and the ground truth (Hinton et al., 2015). It helps with decreasing the overfitting of the victim model (i.e., $\mathcal{W}$). Consequently, $\mathcal{A}$ might be able to remove the watermark and reproduce the core model functionality.

**Model Fine-Tuning**  Fine-tuning (Yosinski et al., 2014) is one of the most commonly used attacks to remove the watermark since it is computationally inexpensive and does not compromise the model's core functionality much. To test GENIE against this attack, we use half of $\mathcal{D}_{test}$ for fine-tuning and evaluate the fine-tuned model's (i.e., $\mathcal{M}_{adv}$) performance with the other half. Through extensive experimentation and analysis, we determined that limiting the training process to 50 epochs serves as an optimal strategy (i.e., to avoid the risk of overfitting $\mathcal{M}_{adv}$ on the subset of $\mathcal{D}_{test}$ used for fine-tuning). We evaluate GENIE against 4 variations of fine-tuning. These can be classified into two broad categories (Yosinski et al., 2014):

**Last layer fine-tuning**: This fine-tuning procedure updates the weights of only the last layer of the target model. It can be done in the following two ways.

1. Fine-Tune Last Layer (**FTLL**): Freezing the weights of the target model, updating the weights of its last layer only during fine-tuning.
2. Re-Train Last Layer (**RTLL**): Freezing the weights of the target model, re-initializing the weights of only its last layer, and then fine-tuning it.

**All layers fine-tuning**: This fine-tuning procedure updates weights of all the layers of the target model. It is a stronger setting compared to the last layer fine-tuning method as all the weights are updated, which makes it tougher to retain the watermark. It can be done in the following two ways.

1. Fine-Tune All Layers (**FTAL**): Updating weights of all the layers of the target model during fine-tuning.
2. Re-Train All Layers (**RTAL**): Re-initializing the weights of target model's last layer, updating weights of all its layers during fine-tuning.

FTLL is considered the weakest attack because it has the least capacity to modify the core GNN layers responsible for learning the watermark. Conversely, RTAL is considered the toughest attack because it enables complete fine-tuning of all model layers, providing the highest flexibility to potentially overwrite or distort the watermark embedded across multiple layers.

**Model Compression** It is a technique to reduce the size and complexity of a DNN, thereby making it more efficient and easily deployable. Compressing the model can inadvertently or otherwise act as an attack against the watermark. Thus, we test GENIE's robustness with following two model compression techniques:

**Model pruning**: Model or parameter pruning (Han et al., 2015) selects a fraction of weights that have the smallest absolute value and makes them zero. It is a computationally inexpensive watermark removal technique.

**Weight quantization:** It is another model compression technique to reduce the size of a model. It changes the representation of weights to a lower-bit system, thereby saving memory. It is often used to compress large models, e.g., LLMs (Gholami et al., 2022). We follow the weight quantization method (Suraj Subramanian & Zhang, 2022) with bit-size $= 3$ (Tan et al., 2023).

**Fine-Pruning** It is a key defense against a backdoor attack that combines model pruning and fine-tuning. It is more effective than individual pruning or fine-tuning, which makes it difficult for the watermark to survive. We start by pruning a fraction of the smallest absolute weights. Next, we fine-tune the pruned model with half of $\mathcal{D}_{test}$ and evaluate the pruned+fine-tuned model (i.e., $\mathcal{M}_{adv}$) with the other half. We perform an exhaustive evaluation with pruning fractions ranging from 0.2-0.8 at a step size of 0.2, which is followed by one of the four types of model fine-tuning (i.e., FTLL, RTLL, FTAL, RTAL). Our rigorous experiments aim to provide a holistic understanding of GENIE's robustness against the fine-pruning technique.

## B Statistical assurance of non-trivial ownership in Genie

The values of $\mathcal{C}_{wm}$ and $\mathcal{W}_{wm}$ are given for $n = 10$ different $\mathcal{C}$ and $(\mathcal{D}_{wm}, \mathcal{W})$ in Table 10 (for SEAL), Table 11 (for GCN), Table 12 (for GraphSAGE), and Table 13 (for NeoGNN). The corresponding $p$-values are also mentioned. To correct for multiple comparison, appropriate corrections such as Bonferroni or BH correction must be applied. However, we observe that for each dataset and architecture, the $p$-value is observed to be much below the significance level, i.e., $1 - \tau = 0.05$ ($p = 0.000$ in most cases). Therefore, we reject $\mathbf{H}_0$ for each architecture and dataset as described in §4.3.

## C Analysis of DWT

### C.1 Sensitivity Analysis

To evaluate the robustness and stability of DWT, we conduct a sensitivity analysis with respect to the number of initial samples and the bandwidth used for estimating the underlying distributions. The primary

| Dataset | | AUC (%) | | | | | | | | | | p-value |
|---------|---|---|---|---|---|---|---|---|---|---|---|---------|
| C.ele | $\mathcal{C}_{wm}$ | 38.63 | 25.60 | 23.70 | 28.72 | 22.48 | 23.29 | 20.39 | 22.62 | 26.83 | 24.97 | 0.000 |
| | $\mathcal{W}_{wm}$ | 76.58 | 77.19 | 77.32 | 78.91 | 78.72 | 77.69 | 78.24 | 77.39 | 78.24 | 75.94 | |
| USAir | $\mathcal{C}_{wm}$ | 8.00 | 6.97 | 5.88 | 7.10 | 7.61 | 7.92 | 6.92 | 8.72 | 7.72 | 8.79 | 0.000 |
| | $\mathcal{W}_{wm}$ | 94.01 | 94.02 | 94.65 | 93.59 | 93.81 | 95.17 | 95.39 | 94.13 | 94.08 | 94.14 | |
| NS | $\mathcal{C}_{wm}$ | 3.41 | 1.73 | 2.25 | 2.57 | 1.91 | 1.72 | 2.04 | 2.50 | 3.70 | 2.10 | 0.000 |
| | $\mathcal{W}_{wm}$ | 98.66 | 98.71 | 98.75 | 98.68 | 98.65 | 98.73 | 98.69 | 98.72 | 98.73 | 98.73 | |
| Yeast | $\mathcal{C}_{wm}$ | 14.37 | 6.73 | 12.49 | 15.54 | 10.21 | 8.03 | 4.23 | 40.05 | 5.02 | 10.72 | 0.000 |
| | $\mathcal{W}_{wm}$ | 97.50 | 98.02 | 98.09 | 97.75 | 97.83 | 97.21 | 97.47 | 97.15 | 97.87 | 97.96 | |
| Power | $\mathcal{C}_{wm}$ | 13.64 | 19.46 | 18.60 | 12.09 | 15.09 | 12.01 | 12.48 | 12.69 | 29.46 | 12.25 | 0.000 |
| | $\mathcal{W}_{wm}$ | 88.28 | 88.31 | 88.33 | 88.27 | 88.25 | 88.24 | 88.28 | 88.26 | 88.31 | 88.32 | |
| arXiv | $\mathcal{C}_{wm}$ | 7.04 | 2.38 | 3.09 | 2.16 | 3.43 | 4.98 | 7.95 | 6.72 | 2.61 | 5.73 | 0.000 |
| | $\mathcal{W}_{wm}$ | 97.98 | 97.88 | 98.18 | 98.26 | 97.94 | 98.08 | 98.36 | 98.33 | 98.30 | 98.23 | |
| PPI | $\mathcal{C}_{wm}$ | 9.96 | 12.76 | 9.68 | 12.09 | 9.90 | 15.35 | 10.44 | 13.89 | 11.99 | 26.02 | 0.000 |
| | $\mathcal{W}_{wm}$ | 83.81 | 83.81 | 84.51 | 82.78 | 86.25 | 83.82 | 84.94 | 84.23 | 81.93 | 86.81 | |

Table 10: Non-trivial ownership results for SEAL.

| Dataset | | AUC (%) | | | | | | | | | | p-value |
|---------|---|---|---|---|---|---|---|---|---|---|---|---------|
| C.ele | $\mathcal{C}_{wm}$ | 4.00 | 7.82 | 0.00 | 0.00 | 13.85 | 0.44 | 7.14 | 2.04 | 11.77 | 31.36 | 0.000 |
| | $\mathcal{W}_{wm}$ | 100.0 | 99.89 | 100.0 | 100.0 | 100.0 | 99.78 | 100.0 | 100.0 | 99.86 | 98.82 | |
| USAir | $\mathcal{C}_{wm}$ | 31.48 | 19.20 | 13.64 | 16.18 | 20.16 | 13.55 | 33.48 | 14.27 | 13.13 | 29.94 | 0.000 |
| | $\mathcal{W}_{wm}$ | 100.0 | 100.0 | 100.0 | 99.91 | 100.0 | 99.96 | 99.83 | 100.0 | 100.0 | 99.78 | |
| NS | $\mathcal{C}_{wm}$ | 0.00 | 3.40 | 11.76 | 4.33 | 0.00 | 6.93 | 15.22 | 13.57 | 5.25 | 40.74 | 0.002 |
| | $\mathcal{W}_{wm}$ | 82.00 | 94.44 | 100.0 | 99.31 | 98.78 | 97.65 | 97.75 | 98.75 | 98.75 | 97.84 | |
| Yeast | $\mathcal{C}_{wm}$ | 18.34 | 16.53 | 13.33 | 0.00 | 18.93 | 27.81 | 18.34 | 12.88 | 10.06 | 18.80 | 0.000 |
| | $\mathcal{W}_{wm}$ | 100.0 | 99.59 | 100.0 | 100.0 | 100.0 | 99.74 | 100.0 | 100.0 | 100.0 | 100.0 | |
| Power | $\mathcal{C}_{wm}$ | 10.07 | 0.00 | 0.00 | 3.56 | 7.69 | 0.00 | 0.00 | 22.31 | 2.55 | 30.79 | 0.000 |
| | $\mathcal{W}_{wm}$ | 98.96 | 98.00 | 94.44 | 100.0 | 92.90 | 98.44 | 99.22 | 97.52 | 98.47 | 100.0 | |
| arXiv | $\mathcal{C}_{wm}$ | 2.26 | 1.24 | 2.91 | 1.09 | 1.58 | 2.08 | 1.72 | 1.45 | 1.26 | 6.67 | 0.000 |
| | $\mathcal{W}_{wm}$ | 100.0 | 99.99 | 100.0 | 100.0 | 100.0 | 99.99 | 100.0 | 99.98 | 100.0 | 100.0 | |
| PPI | $\mathcal{C}_{wm}$ | 8.07 | 8.96 | 5.78 | 4.48 | 6.94 | 6.18 | 4.17 | 5.28 | 10.85 | 22.14 | 0.000 |
| | $\mathcal{W}_{wm}$ | 100.0 | 100.0 | 100.0 | 99.98 | 99.96 | 100.0 | 100.0 | 100.0 | 99.97 | 99.96 | |

Table 11: Non-trivial ownership results for GCN.

objective is to understand how the sample size and bandwidth, denoted by $s$ and $h$ respectively, influences the mean and variance of the resulting watermark threshold, $t$, even under cases where a large $h$ would yield overlapping probability distributions.

The analysis employs a Monte Carlo simulation framework. For each dataset, we vary (1) the sample size $s$ over values $\{4, 5, \ldots, 10\}$ keeping the bandwidth $h = h_0$ constant; (2) the bandwidth $h$ over values $\{0.5h_0, 0.75h_0, h_0, 1.25h_0, 1.5h_0, 2h_0\}$ keeping $s = 10$ constant. Here, $h_0$ is the bandwidth obtained using

| Dataset | | AUC (%) | | | | | | | | | | *p*-value |
|---------|---|---|---|---|---|---|---|---|---|---|---|---|
| C.ele | $\mathcal{C}_{wm}$ | 7.63 | 3.32 | 9.57 | 12.40 | 2.22 | 25.78 | 7.62 | 12.93 | 13.67 | 15.38 | 0.000 |
| | $\mathcal{W}_{wm}$ | 99.88 | 99.86 | 99.61 | 99.59 | 99.78 | 100.00 | 99.61 | 99.55 | 100.0 | 100.0 | |
| USAir | $\mathcal{C}_{wm}$ | 8.57 | 5.78 | 7.43 | 3.78 | 11.09 | 3.86 | 18.38 | 7.64 | 20.85 | 25.00 | 0.000 |
| | $\mathcal{W}_{wm}$ | 99.90 | 100.0 | 100.0 | 99.96 | 100.0 | 100.0 | 100.0 | 100.0 | 100.0 | 100.0 | |
| NS | $\mathcal{C}_{wm}$ | 9.34 | 16.44 | 19.66 | 7.76 | 28.39 | 11.25 | 18.00 | 21.28 | 17.46 | 25.00 | 0.000 |
| | $\mathcal{W}_{wm}$ | 91.52 | 99.11 | 97.16 | 96.95 | 97.31 | 96.50 | 89.75 | 97.68 | 96.28 | 97.62 | |
| Yeast | $\mathcal{C}_{wm}$ | 0.00 | 18.00 | 20.14 | 31.00 | 23.47 | 7.96 | 0.00 | 18.37 | 12.24 | 40.62 | 0.007 |
| | $\mathcal{W}_{wm}$ | 100.0 | 100.0 | 100.0 | 100.0 | 100.0 | 100.0 | 100.0 | 100.0 | 100.0 | 100.0 | |
| Power | $\mathcal{C}_{wm}$ | 0.00 | 2.00 | 30.56 | 2.47 | 3.56 | 0.00 | 0.00 | 15.00 | 12.00 | 25.00 | 0.000 |
| | $\mathcal{W}_{wm}$ | 97.53 | 92.00 | 97.57 | 96.30 | 94.44 | 97.53 | 96.28 | 93.50 | 98.00 | 92.97 | |
| arXiv | $\mathcal{C}_{wm}$ | 0.00 | 2.00 | 30.56 | 2.47 | 3.56 | 0.00 | 0.00 | 15.00 | 12.00 | 25.00 | 0.000 |
| | $\mathcal{W}_{wm}$ | 97.53 | 92.00 | 97.57 | 96.30 | 94.44 | 97.53 | 96.28 | 93.50 | 98.00 | 92.97 | |
| PPI | $\mathcal{C}_{wm}$ | 23.12 | 12.98 | 16.74 | 3.12 | 9.58 | 4.92 | 7.46 | 4.39 | 17.12 | 25.00 | 0.000 |
| | $\mathcal{W}_{wm}$ | 100.0 | 100.0 | 99.90 | 99.97 | 99.89 | 100.0 | 99.97 | 100.0 | 100.0 | 99.83 | |

Table 12: Non-trivial ownership results for GraphSAGE.

| Dataset | | AUC (%) | | | | | | | | | | *p*-value |
|---------|---|---|---|---|---|---|---|---|---|---|---|---|
| Celegans | $\mathcal{C}_{wm}$ | 20.31 | 28.12 | 23.44 | 12.50 | 25.00 | 14.06 | 18.75 | 23.44 | 23.44 | 15.62 | 0.000 |
| | $\mathcal{W}_{wm}$ | 100.0 | 100.0 | 100.0 | 100.0 | 100.0 | 100.0 | 100.0 | 100.0 | 100.0 | 100.0 | |
| USAir | $\mathcal{C}_{wm}$ | 11.33 | 12.60 | 10.16 | 12.89 | 11.23 | 11.33 | 10.64 | 14.84 | 11.91 | 9.18 | 0.000 |
| | $\mathcal{W}_{wm}$ | 100.0 | 100.0 | 100.0 | 100.0 | 100.0 | 100.0 | 100.0 | 100.0 | 100.0 | 100.0 | |
| NS | $\mathcal{C}_{wm}$ | 23.11 | 20.22 | 28.89 | 26.44 | 31.78 | 26.44 | 17.33 | 28.89 | 23.11 | 31.78 | 0.000 |
| | $\mathcal{W}_{wm}$ | 100.0 | 100.0 | 100.0 | 100.0 | 100.0 | 100.0 | 100.0 | 100.0 | 100.0 | 100.0 | |
| Yeast | $\mathcal{C}_{wm}$ | 5.33 | 7.10 | 8.28 | 9.47 | 9.47 | 6.80 | 8.88 | 8.28 | 5.62 | 9.47 | 0.000 |
| | $\mathcal{W}_{wm}$ | 100.0 | 100.0 | 100.0 | 100.0 | 100.0 | 100.0 | 100.0 | 100.0 | 100.0 | 100.0 | |
| Power | $\mathcal{C}_{wm}$ | 50.00 | 45.00 | 45.00 | 50.00 | 50.00 | 50.00 | 50.00 | 50.00 | 50.00 | 45.00 | 0.000 |
| | $\mathcal{W}_{wm}$ | 100.0 | 100.0 | 100.0 | 97.00 | 100.0 | 100.0 | 100.0 | 100.0 | 100.0 | 100.0 | |
| arXiv | $\mathcal{C}_{wm}$ | 8.38 | 1.17 | 9.42 | 7.64 | 1.25 | 8.15 | 9.01 | 8.20 | 7.05 | 9.86 | 0.000 |
| | $\mathcal{W}_{wm}$ | 97.34 | 94.03 | 93.64 | 90.38 | 87.73 | 96.19 | 97.44 | 99.97 | 98.70 | 92.44 | |
| PPI | $\mathcal{C}_{wm}$ | 24.52 | 2.85 | 15.15 | 11.11 | 10.47 | 16.25 | 17.95 | 6.52 | 10.19 | 12.81 | 0.000 |
| | $\mathcal{W}_{wm}$ | 100.0 | 99.91 | 97.43 | 97.61 | 95.32 | 94.63 | 97.52 | 97.06 | 100.0 | 96.97 | |

Table 13: Non-trivial ownership results for NeoGNN.

Silverman's rule of thumb (Silverman, 2018). For fixed value of $s$ and $t$, we perform a simulation consisting of 1000 independent trials. The procedure for each trial is as follows:

1. **Subsampling:** We randomly draw, without replacement, a subsample of $s$ data points from the complete set of scores obtained from clean models ($\mathcal{C}$). Concurrently, we draw another subsample of size $s$ from the complete set of scores from watermarked models ($\mathcal{W}$).

| $s$ | C.ele | USAir | NS | Yeast | Power | arXiv | PPI |
|---|---|---|---|---|---|---|---|
| 4 | $38.46 \pm 21.31$ | $50.08 \pm 6.28$ | $48.65 \pm 29.07$ | $36.63 \pm 7.38$ | $41.06 \pm 20.95$ | $7.72 \pm 4.16$ | $24.31 \pm 12.91$ |
| 5 | $38.04 \pm 18.95$ | $49.91 \pm 4.02$ | $48.65 \pm 26.04$ | $37.57 \pm 7.34$ | $40.80 \pm 18.40$ | $7.69 \pm 3.79$ | $23.65 \pm 12.06$ |
| 6 | $38.98 \pm 17.07$ | $49.43 \pm 3.16$ | $48.60 \pm 23.60$ | $39.37 \pm 6.55$ | $40.96 \pm 16.42$ | $7.51 \pm 3.52$ | $23.97 \pm 11.61$ |
| 7 | $38.96 \pm 15.32$ | $49.46 \pm 2.47$ | $48.01 \pm 21.35$ | $40.50 \pm 5.67$ | $42.36 \pm 13.71$ | $7.31 \pm 3.30$ | $23.51 \pm 11.06$ |
| 8 | $38.61 \pm 14.30$ | $49.31 \pm 1.67$ | $46.58 \pm 19.98$ | $42.02 \pm 3.91$ | $45.53 \pm 7.60$ | $7.09 \pm 3.13$ | $23.38 \pm 10.65$ |
| 9 | $44.79 \pm 11.33$ | $49.10 \pm 1.38$ | $55.60 \pm 16.93$ | $41.80 \pm 2.60$ | $48.37 \pm 6.62$ | $8.67 \pm 2.53$ | $28.18 \pm 7.89$ |
| 10 | $49.67 \pm 0.79$ | $49.10 \pm 0.50$ | $63.63 \pm 0.92$ | $41.64 \pm 0.56$ | $51.24 \pm 0.85$ | $9.84 \pm 0.13$ | $32.24 \pm 0.43$ |
| $t$ | 50.65 | 49.69 | 64.82 | 42.35 | 52.29 | 10.00 | 32.77 |

Table 14: Sensitivity analysis of DWT to the number of samples $s$. $t$ is the reported threshold.

| $h$ | C.ele | USAir | NS | Yeast | Power | arXiv | PPI |
|---|---|---|---|---|---|---|---|
| $0.5h_0$ | $40.24 \pm 0.56$ | $40.99 \pm 0.44$ | $51.88 \pm 0.67$ | $34.49 \pm 0.43$ | $40.68 \pm 0.60$ | $8.21 \pm 0.09$ | $27.05 \pm 0.31$ |
| $0.75h_0$ | $44.66 \pm 0.80$ | $44.81 \pm 0.61$ | $57.34 \pm 1.04$ | $37.83 \pm 0.62$ | $45.63 \pm 0.93$ | $8.98 \pm 0.14$ | $29.47 \pm 0.45$ |
| $h_0$ | $49.11 \pm 1.10$ | $48.64 \pm 0.79$ | $62.90 \pm 1.38$ | $41.22 \pm 0.80$ | $50.52 \pm 1.22$ | $9.73 \pm 0.19$ | $31.90 \pm 0.61$ |
| $1.25h_0$ | $53.56 \pm 1.35$ | $52.55 \pm 0.89$ | $68.08 \pm 1.66$ | $44.54 \pm 1.02$ | $55.56 \pm 1.42$ | $10.50 \pm 0.23$ | $34.35 \pm 0.77$ |
| $1.5h_0$ | $57.85 \pm 1.68$ | $56.46 \pm 1.01$ | $68.47 \pm 1.10$ | $47.92 \pm 1.21$ | $60.66 \pm 1.72$ | $11.27 \pm 0.28$ | $36.77 \pm 0.89$ |
| $2h_0$ | $66.93 \pm 2.09$ | $64.37 \pm 1.25$ | $68.33 \pm 0.95$ | $54.84 \pm 1.46$ | $70.69 \pm 2.21$ | $12.79 \pm 0.37$ | $41.68 \pm 1.21$ |
| $t$ | 50.65 | 49.69 | 64.82 | 42.35 | 52.29 | 10.00 | 32.77 |

Table 15: Sensitivity analysis of DWT to bandwidth $h$. $t$ is the reported threshold.

2. **Threshold Estimation with $s$:** The DWT procedure is executed using the two smaller subsamples of size $s$, with threshold value, $t_i$, for the $i$-th trial.

3. **Threshold Estimation with $h$:** The DWT procedure is executed using bandwidth $h$, with threshold value, $t_i$, for the $i$-th trial.

After executing all 1000 trials for a given $s$ or $h$, we obtain 1000 threshold values, $\{t_1, t_2, \ldots, t_{1000}\}$. From this distribution, we compute the sample mean ($\bar{t}$) and sample standard deviation ($\sigma_t$) to quantify the expected threshold and its variability. This entire process is repeated for each value of $s \in \{4, 5, \ldots, 10\}$ and $h \in \{0.5h_0,\ 0.75h_0,\ h_0,\ 1.25h_0,\ 1.5h_0,\ 2h_0\}$ for every dataset under evaluation. Table 14, 15 show the results for GCN. In both the tables, we observe the reported threshold $t$ to be more than the mean threshold in most cases, indicating the reported thresholds avoid large FPR. We also see $t$ less when an overly smooth $h \geq 1.5h_0$ is used to yield overlapping distributions, indicating the reported thresholds avoid large FNR as well.

## C.2   Theoretical Analysis

We elaborate on the proof sketch of the theorem given in §4.3, adapted from Eypasch et al. (1995).

*Proof.* We first note that

$$\Pr(X_j = 0) \; = \; (1 - p)^n. \tag{3}$$

Hence, observing $X_j = 0$ for all $j = 1, \ldots, m$ yields

$$\Pr(X_1 = 0, \ldots, X_m = 0) \; = \; \big((1-p)^n\big)^m \; = \; (1-p)^{mn}. \tag{4}$$

If $p \geq \frac{1}{n}$, then using the inequality, $e^x \geq 1 + x \ \forall \ x \in \mathbb{R}$ we get,

$$(1 - p) \; \leq \; e^{-p} \; \leq \; e^{-\frac{1}{n}}, \tag{5}$$

implying

$$(1 - p)^{mn} \; \leq \; \exp\big(-mnp\big) \; \leq \; e^{-m}. \tag{6}$$

Thus,

$$\Pr\left(\text{all zeros} \mid p \geq 1/n\right) \leq e^{-m}. \tag{7}$$

If we choose $m$ such that

$$e^{-m} \leq 1 - \gamma \quad \Longleftrightarrow \quad m \geq -\ln(1-\gamma), \tag{8}$$

then the probability of observing zero events in all $m$ blocks *despite* $p \geq 1/n$ is at most $1 - \gamma$. Equivalently, whenever we *do* observe zero events in all $m$ blocks, it follows with confidence at least $\gamma$ that $p < 1/n$. $\qquad\square$

## D  Ownership Demonstration

We now outline the process for $\mathcal{O}$ to demonstrate her ownership over $\mathcal{A}$'s model (i.e., $\mathcal{M}_{adv}$). OD uses $\mathcal{J}$ briefly outlined in §3. GENIE has a two-step OD procedure that involves a *model registration* step and a *dispute resolution* step.

**Model registration:** As a preemptive step of OD, $\mathcal{O}$ first sends $\mathcal{G}$ to $\mathcal{J}$ to procure $\mathcal{D}_{wm}$, addressing the problem of malicious plaintiff outlined in (Liu et al., 2024). $\mathcal{J}$ then writes the cryptographic hash of $\mathcal{D}_{wm}$ onto a time-stamped public bulletin board (e.g., blockchain) to provide the proof of anteriority in case of a dispute. We call this preemptive step model registration since it is analogous to patent registration common in the protection of IP rights (Waheed et al., 2023; Park, 2008). The procured $\mathcal{D}_{wm}$ will then be used for embedding the watermark into $\mathcal{M}$, i.e., train an untrained $\mathcal{M}$ to be $\mathcal{W}$ by $\mathcal{O}$.

**Dispute resolution:** When a dispute arises, OD involves the following steps: (1) $\mathcal{O}$ accuses $\mathcal{A}$ of plagiarizing her model $\mathcal{W}$; (2) $\mathcal{A}$ sends $\mathcal{M}_{adv}$ to $\mathcal{J}$ for an evaluation; (3) $\mathcal{O}$ sends $\mathcal{D}_{wm}$ and the hashes of all the files. Here, the hashes are sent via a secure communication channel to ensure that the files are not tampered with; (4) $\mathcal{J}$ runs a check on the hashes of the files sent. Next, $\mathcal{J}$ checks the record of $\mathcal{D}_{wm}$ in the public bulletin board. If a matching record is found, $\mathcal{J}$ first calculates the watermark threshold $t$, and evaluates $\mathcal{M}_{adv}$ on $\mathcal{D}_{wm}$ to get $\mathcal{W}_{wm}$. The OD ends with a comparison of $\mathcal{W}_{wm}$ against $t$, settling the dispute between $\mathcal{O}$ and $\mathcal{A}$ with a just verdict. On the other hand, if a record is not found, the dispute resolves in $\mathcal{O}$'s defeat.

## E  Time and Space Complexity Analysis

**Node-Representation Based Methods.** *Time Complexity:* The total time is the sum of the times for each step in the watermark generation process.

1. **Node Sampling:** Sampling $|\mathcal{S}|$ nodes from $|\mathcal{V}|$ nodes uniformly at random takes $O(|\mathcal{S}|) = O(\alpha_{nr}|\mathcal{V}|)$ time.

2. **Watermark Graph Construction:** Constructing the watermark graph $\mathcal{G}_{wm}$ involves creating the complement of the subgraph induced by $\mathcal{S}$, denoted $\mathcal{G}_{\mathcal{S}}$. This requires iterating through all possible pairs of nodes in $\mathcal{S}$ to determine the edges in the complement $\overline{\mathcal{E}}_{\mathcal{S}}$. This step has a time complexity of $O(|\mathcal{S}|^2) = O((\alpha_{nr}|\mathcal{V}|)^2)$.

3. **Feature Matrix Modification:** For each of the $|\mathcal{S}|$ nodes, the $d$-dimensional feature vector is replaced with the watermark vector $\mathbf{w}$. This operation takes $O(|\mathcal{S}| \cdot d) = O(\alpha_{nr}|\mathcal{V}|d)$ time.

Combining these steps, the dominant terms determine the overall time complexity. Therefore, the total time complexity is $O((\alpha_{nr}|\mathcal{V}|)^2 + \alpha_{nr}|\mathcal{V}|d)$.

*Space Complexity:* The space complexity is determined by the storage requirements for the generated watermark dataset $\mathcal{D}_{wm} = (\mathbf{E}_{wm}, \mathbf{A}_{wm}, \mathbf{X}_{wm}, \mathbf{y}_{wm})$.

1. **Adjacency Matrix $\mathbf{A}_{wm}$:** Storing the adjacency matrix for the entire graph $\mathcal{G}_{wm}$ requires $O(|\mathcal{V}|^2)$ space.

2. **Feature Matrix $\mathbf{X}_{wm}$:** Storing the modified node feature matrix for all nodes requires $O(|\mathcal{V}|d)$ space.

3. **Edge Index $\mathbf{E}_{wm}$:** The edge index for the watermarked portion contains edges from $\mathcal{E}_{\mathcal{S}} \cup \overline{\mathcal{E}}_{\mathcal{S}}$. The number of these edges is bounded by $O(|\mathcal{S}|^2)$. This is subsumed by the $O(|\mathcal{V}|^2)$ term for the full adjacency matrix.

*Proof.* The total space complexity is dominated by the storage of the full graph's adjacency and feature matrices, resulting in $O(|\mathcal{V}|^2 + |\mathcal{V}|d)$. □

*Proof.* The proof is divided into analyzing the time and space requirements.

**Time Complexity:** The process involves sampling subgraphs and modifying the features of all nodes within them.

1. **Subgraph Sampling:** Sampling $s = \lceil \alpha_{sg}T \rceil$ subgraphs from the training set of size $T$ takes $O(s) = O(\alpha_{sg}T)$ time.

2. **Feature Matrix Modification:** This is the most computationally intensive step. For each of the $s$ sampled subgraphs, we iterate through its $N_{sub}$ nodes and replace their $d$-dimensional feature vectors with the watermark vector $\mathbf{w}$. The total time for this operation is $O(s \cdot N_{sub} \cdot d) = O(\alpha_{sg}T \cdot N_{sub} \cdot d)$.

3. **Label Inversion:** Inverting the labels for the $s$ subgraphs takes $O(s)$ time.

The feature modification step is the dominant factor. Thus, the total time complexity is $O(\alpha_{sg}T \cdot N_{sub} \cdot d)$.

**Space Complexity:** The space complexity is determined by the storage needed for the watermark dataset $\mathcal{D}_{wm}$, which consists of the $s$ modified subgraphs.

1. For each of the $s$ modified subgraphs, we need to store its structure and node features.

2. **Node Features:** Storing the features for $N_{sub}$ nodes, each of dimension $d$, requires $O(N_{sub} \cdot d)$ space per subgraph.

3. **Subgraph Structure:** Storing the adjacency information for a subgraph of $N_{sub}$ nodes (e.g., as an adjacency matrix) requires $O(N_{sub}^2)$ space per subgraph.

The total space required is the space per subgraph multiplied by the number of subgraphs $s$. Therefore, the total space complexity is $O(s \cdot (N_{sub}d + N_{sub}^2)) = O(\alpha_{sg}T(N_{sub}d + N_{sub}^2))$. □

# F  Additional Results

## F.1  Inductive Link Prediction results

Inductive link prediction is useful in case where predictions need to be made on unseen nodes that are not present during the training time. Accordingly, the threat model for watermarking inductive link prediction also differs significantly from the transductive setting. We summarize the difference between the two settings in Table 16.

We watermark the state-of-the-art inductive link prediction method LEAP (Samy et al., 2024) using GENIE, keeping all hyperparameters identical to those used in LEAP. On Wikipedia (Rozemberczki et al., 2021a) and PubMed (Yang et al., 2016), at a 3% watermarking rate (Table 17), GENIE attains high $\mathcal{W}_{wm}$ of 97.25% and 100.00% while having a drop from $\mathcal{C}_{test}$ to $\mathcal{W}_{test}$ by less than 2 percentage points (a 0.55% gain on Wikipedia and a 0.39% drop on PubMed).

| Aspect | Transductive | Inductive |
|---|---|---|
| Graph scope | Fixed graph at train and test time. | New nodes or graphs appear at test time. |
| Adversary observability | Adversary can observe or reconstruct large parts of the training (target) graph. | Adversary lacks access to training graph. |
| Attack surface | Query-based graph reconstruction, fine-tuning, pruning on same graph. | Model extraction and retraining on different graphs. |
| Watermark exposure | Model can overfit to triggers; higher watermark verification easier. | Triggers must generalize; higher watermark verification is harder. |
| Threat model implication | Stronger adversary with high graph visibility. | Weaker adversary due to limited data overlap. |

Table 16: Threat model differences between transductive and inductive link prediction settings

| Metric | Wikipedia (%) | PubMed (%) |
|---|---|---|
| $\mathcal{C}_{test}$ | 89.50 | 94.90 |
| $\mathcal{W}_{test}$ | 90.05 | 94.51 |
| $\mathcal{W}_{wm}$ | 97.25 | 100.00 |

Table 17: AUC metrics on Wikipedia and PubMed. Watermarking rate is 3%

## F.2 Effect of Watermark Vector Distribution

We analyze the effect of watermark vector distribution using GCN with watermarking rates as mentioned in A.3 on USAir, Celegans and NS datasets in 18. We find that the watermark vector distribution has almost no effect on the performance GENIE making it robust to multiple distributions.

| Dataset | | Watermark vector distribution | | | | |
|---|---|---|---|---|---|---|
| | | Normal | Uniform | Poisson | Exponential | Bernoulli |
| USAir | $\mathcal{W}_{test}$ | 88.34 | 89.34 | 88.83 | 87.29 | 87.90 |
| | $\mathcal{W}_{wm}$ | 100 | 100 | 100 | 100 | 100 |
| C.ele | $\mathcal{W}_{test}$ | 86.93 | 88.15 | 87.17 | 88.48 | 88.39 |
| | $\mathcal{W}_{wm}$ | 100 | 100 | 100 | 100 | 100 |
| NS | $\mathcal{W}_{test}$ | 96.59 | 96.02 | 97.10 | 95.78 | 95.47 |
| | $\mathcal{W}_{wm}$ | 99.77 | 99.77 | 99.77 | 99.77 | 99.77 |

Table 18: Impact of Watermark vector distribution

## F.3 GCN

### F.3.1 White-Box Attacks at different % (Robustness)

**Pruning** Table 19 shows the results of GCN watermarked using GENIE under Pruning attack from 20-80% pruning percentages.

| Dataset | | Prune Percentage (%) | | | | |
|---|---|---|---|---|---|---|
| | | 0 | 20 | 40 | 60 | 80 |
| C.ele | $\mathcal{W}_{test}$ | 86.93 | 86.97 | 86.31 | 83.92 | 75.43 |
| | $\mathcal{W}_{wm}$ | 100 | 100 | 100 | 100 | 70.31 |
| USAir | $\mathcal{W}_{test}$ | 88.34 | 88.98 | 89.57 | 89.21 | 78.50 |
| | $\mathcal{W}_{wm}$ | 100 | 100 | 100 | 92.77 | 71.28 |
| NS | $\mathcal{W}_{test}$ | 96.59 | 96.25 | 96.01 | 96.08 | 91.31 |
| | $\mathcal{W}_{wm}$ | 99.77 | 99.77 | 99.77 | 96.66 | 85.55 |
| Yeast | $\mathcal{W}_{test}$ | 91.46 | 91.27 | 89.97 | 85.71 | 80.50 |
| | $\mathcal{W}_{wm}$ | 100 | 100 | 100 | 100 | 74.55 |
| Power | $\mathcal{W}_{test}$ | 98.92 | 99.00 | 99.20 | 98.32 | 92.43 |
| | $\mathcal{W}_{wm}$ | 99.00 | 99.00 | 95.00 | 74.00 | 55.00 |
| arXiv | $\mathcal{W}_{test}$ | 98.13 | 98.08 | 97.911 | 94.79 | 84.37 |
| | $\mathcal{W}_{wm}$ | 100 | 100 | 99.98 | 83.09 | 41.95 |
| PPI | $\mathcal{W}_{test}$ | 94.67 | 94.60 | 94.05 | 92.86 | 90.06 |
| | $\mathcal{W}_{wm}$ | 100 | 100 | 100 | 78.87 | ~~31.22~~ |

Table 19: Impact of model pruning.

**Fine-pruning**  Tables 20-23 present results for different fine-pruning tests for the GCN model.

| Dataset | | Prune Percentage (%) | | | | |
|---|---|---|---|---|---|---|
| | | 0 | 20 | 40 | 60 | 80 |
| C.ele | $\mathcal{W}_{test}$ | 86.93 | 82.01 | 80.80 | 78.54 | 82.57 |
| | $\mathcal{W}_{wm}$ | 100 | 93.75 | 90.62 | 81.25 | 79.68 |
| USAir | $\mathcal{W}_{test}$ | 88.34 | 89.15 | 88.85 | 88.30 | 87.93 |
| | $\mathcal{W}_{wm}$ | 100 | 91.21 | 90.42 | 87.50 | 69.82 |
| NS | $\mathcal{W}_{test}$ | 96.59 | 98.49 | 98.22 | 97.55 | 97.26 |
| | $\mathcal{W}_{wm}$ | 99.77 | 99.77 | 98.88 | 97.55 | 93.11 |
| Yeast | $\mathcal{W}_{test}$ | 91.46 | 91.52 | 91.41 | 89.90 | 86.64 |
| | $\mathcal{W}_{wm}$ | 100 | 91.71 | 90.53 | 85.20 | 92.30 |
| Power | $\mathcal{W}_{test}$ | 98.92 | 99.38 | 99.30 | 98.91 | 98.04 |
| | $\mathcal{W}_{wm}$ | 99.00 | 99.00 | 95.00 | 87.00 | 72.00 |
| arXiv | $\mathcal{W}_{test}$ | 98.13 | 98.54 | 98.41 | 98.07 | 96.74 |
| | $\mathcal{W}_{wm}$ | 100 | 86.51 | 81.81 | 59.24 | 30.07 |
| PPI | $\mathcal{W}_{test}$ | 94.67 | 94.92 | 94.73 | 94.63 | 93.41 |
| | $\mathcal{W}_{wm}$ | 100 | 96.32 | 88.52 | 79.43 | 51.97 |

Table 20: Impact of pruning + FTLL.

| Dataset | | Prune Percentage (%) | | | | |
|---|---|---|---|---|---|---|
| | | 0 | 20 | 40 | 60 | 80 |
| C.ele | $\mathcal{W}_{test}$ | 86.93 | 70.65 | 71.04 | 73.56 | 79.64 |
| | $\mathcal{W}_{wm}$ | 100 | 59.37 | 57.81 | 68.75 | 84.37 |
| USAir | $\mathcal{W}_{test}$ | 88.34 | 87.57 | 87.50 | 88.09 | 86.59 |
| | $\mathcal{W}_{wm}$ | 100 | 82.12 | 82.81 | 79.58 | 61.62 |
| NS | $\mathcal{W}_{test}$ | 96.59 | 98.43 | 98.18 | 97.74 | 96.72 |
| | $\mathcal{W}_{wm}$ | 99.77 | 96.66 | 97.55 | 96.22 | 94.44 |
| Yeast | $\mathcal{W}_{test}$ | 91.46 | 90.72 | 90.19 | 88.87 | 86.24 |
| | $\mathcal{W}_{wm}$ | 100 | 88.16 | 88.16 | 77.51 | 84.61 |
| Power | $\mathcal{W}_{test}$ | 98.92 | 99.24 | 99.16 | 98.73 | 97.40 |
| | $\mathcal{W}_{wm}$ | 99.00 | 99.00 | 93.00 | 91.99 | 71.00 |
| arXiv | $\mathcal{W}_{test}$ | 98.13 | 98.54 | 98.41 | 98.07 | 96.74 |
| | $\mathcal{W}_{wm}$ | 100 | 44.95 | 47.04 | 48.31 | 39.69 |
| PPI | $\mathcal{W}_{test}$ | 94.67 | 94.21 | 94.04 | 93.54 | 92.78 |
| | $\mathcal{W}_{wm}$ | 100 | 59.41 | 51.42 | 53.16 | 46.28 |

Table 21: Impact of pruning + RTLL.

| Dataset | | Prune Percentage (%) | | | | |
|---|---|---|---|---|---|---|
| | | 0 | 20 | 40 | 60 | 80 |
| C.ele | $\mathcal{W}_{test}$ | 86.93 | 75.73 | 77.27 | 75.46 | 74.68 |
| | $\mathcal{W}_{wm}$ | 100 | 71.87 | 60.93 | 78.12 | 62.50 |
| USAir | $\mathcal{W}_{test}$ | 88.34 | 86.35 | 85.96 | 86.94 | 85.23 |
| | $\mathcal{W}_{wm}$ | 100 | 81.49 | 73.19 | 76.31 | 63.28 |
| NS | $\mathcal{W}_{test}$ | 96.59 | 91.90 | 89.33 | 88.09 | 87.84 |
| | $\mathcal{W}_{wm}$ | 99.77 | 89.11 | 84.66 | 88.66 | 92.22 |
| Yeast | $\mathcal{W}_{test}$ | 91.46 | 90.48 | 90.06 | 89.43 | 87.45 |
| | $\mathcal{W}_{wm}$ | 100 | 100 | 100 | 99.40 | 81.65 |
| Power | $\mathcal{W}_{test}$ | 98.92 | 97.36 | 97.39 | 97.69 | 96.97 |
| | $\mathcal{W}_{wm}$ | 99.00 | 88.00 | 87.00 | 84.00 | 67.99 |
| arXiv | $\mathcal{W}_{test}$ | 98.13 | 98.78 | 98.79 | 98.80 | 98.48 |
| | $\mathcal{W}_{wm}$ | 100 | 54.45 | 48.18 | 33.05 | 17.00 |
| PPI | $\mathcal{W}_{test}$ | 94.67 | 94.02 | 94.04 | 93.93 | 93.87 |
| | $\mathcal{W}_{wm}$ | 100 | 78.60 | 75.39 | 66.20 | 45.36 |

Table 22: Impact of pruning + FTAL.

| Dataset | | Prune Percentage (%) | | | | |
|---|---|---|---|---|---|---|
| | | 0 | 20 | 40 | 60 | 80 |
| C.ele | $\mathcal{W}_{test}$ | 86.93 | 69.11 | 66.34 | 72.06 | 68.28 |
| | $\mathcal{W}_{wm}$ | 100 | 68.75 | 68.75 | 70.31 | 75.00 |
| USAir | $\mathcal{W}_{test}$ | 88.34 | 85.46 | 84.94 | 83.26 | 84.50 |
| | $\mathcal{W}_{wm}$ | 100 | 63.76 | 65.52 | 62.79 | 47.75 |
| NS | $\mathcal{W}_{test}$ | 96.59 | 76.49 | 79.69 | 74.29 | 74.74 |
| | $\mathcal{W}_{wm}$ | 99.77 | 78.00 | 85.11 | 78.88 | 78.88 |
| Yeast | $\mathcal{W}_{test}$ | 91.46 | 85.70 | 85.47 | 84.50 | 82.89 |
| | $\mathcal{W}_{wm}$ | 100 | 84.02 | 92.89 | 89.94 | 63.31 |
| Power | $\mathcal{W}_{test}$ | 98.92 | 92.87 | 92.33 | 91.92 | 90.02 |
| | $\mathcal{W}_{wm}$ | 99.00 | 63.00 | 64.99 | 63.00 | 49.99 |
| arXiv | $\mathcal{W}_{test}$ | 98.13 | 98.02 | 98.01 | 97.61 | 95.82 |
| | $\mathcal{W}_{wm}$ | 100 | 20.67 | 19.46 | 17.99 | 13.93 |
| PPI | $\mathcal{W}_{test}$ | 94.67 | 92.92 | 92.79 | 92.45 | 91.88 |
| | $\mathcal{W}_{wm}$ | 100 | 42.51 | 44.81 | 46.74 | 27.08 |

Table 23: Impact of pruning + RTAL.

### F.3.2  Non-Ownership Piracy

GENIE injects watermark into an untrained model while $\mathcal{A}$ has access to only $\mathcal{W}$. In a real-world setting, $\mathcal{A}$ can still generate her own pirated trigger set and train stolen $\mathcal{W}$ to obtain $\mathcal{M}_{adv}$ (generally called a **pirated model**). Given that training on just the pirated trigger set might lead to decrease in $\mathcal{W}_{test}$, $\mathcal{A}$ would want to identify an optimal number of epochs for training with the pirated trigger set such that $\mathcal{M}_{adv}$ has high AUC on both $\mathcal{D}_{test}$ and the pirated trigger set. Figure 4 shows the variations in $\mathcal{M}_{adv}$'s performance on $\mathcal{D}_{wm}$, $\mathcal{D}_{test}$, and pirated trigger set during pirated watermark embedding process across different numbers of epochs for GCN over NS dataset; cf. Figure 5 for other datasets.

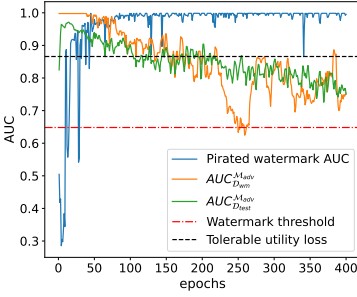

Figure 4: An example of $\mathcal{M}_{adv}$'s performance trajectory on $\mathcal{D}_{wm}$, $\mathcal{D}_{test}$, and pirated trigger set during embedding of pirated watermark across training. $AUC_{\mathcal{D}_{wm}}^{\mathcal{M}_{adv}}$ and $AUC_{\mathcal{D}_{test}}^{\mathcal{M}_{adv}}$ denote $\mathcal{W}_{wm}$ and $\mathcal{W}_{test}$, respectively.

We see that $\mathcal{M}_{adv}$ performs well on $\mathcal{D}_{wm}$, $\mathcal{D}_{test}$, as well as on pirated trigger sets around $20^{th}$ epoch. If $\mathcal{O}$ challenges $\mathcal{A}$ to present her model at this point, $\mathcal{M}_{adv}$ will contain $\mathcal{A}$'s pirated watermark as well as $\mathcal{O}$'s watermark. However, $\mathcal{O}$ can present $\mathcal{W}$ containing only her watermark. Thus, identifying the true owner will be easy in such a dispute. We further observe that around $250^{th}$ epoch, $AUC_{\mathcal{D}_{wm}}^{\mathcal{M}_{adv}}$ drops below the watermark threshold (cf. Table 1), but $\mathcal{W}_{test}$ is below the tolerable utility loss (i.e., up to 10%; following the definition of failure in §5.4) $\mathcal{A}$ is willing to tolerate. Even if $\mathcal{A}$ chooses to train for even higher epochs while embedding the pirated watermark, $\mathcal{M}_{adv}$ continues to lose its utility; rendering the $\mathcal{M}_{adv}$ useless. Hence, we take the liberty to claim that GENIE **is robust against piracy attacks** (i.e., $\mathcal{A}$ cannot fraudulently claim ownership or fabricate watermark over a pirated model). Now, we present the results for the other datasets in Figure 5.

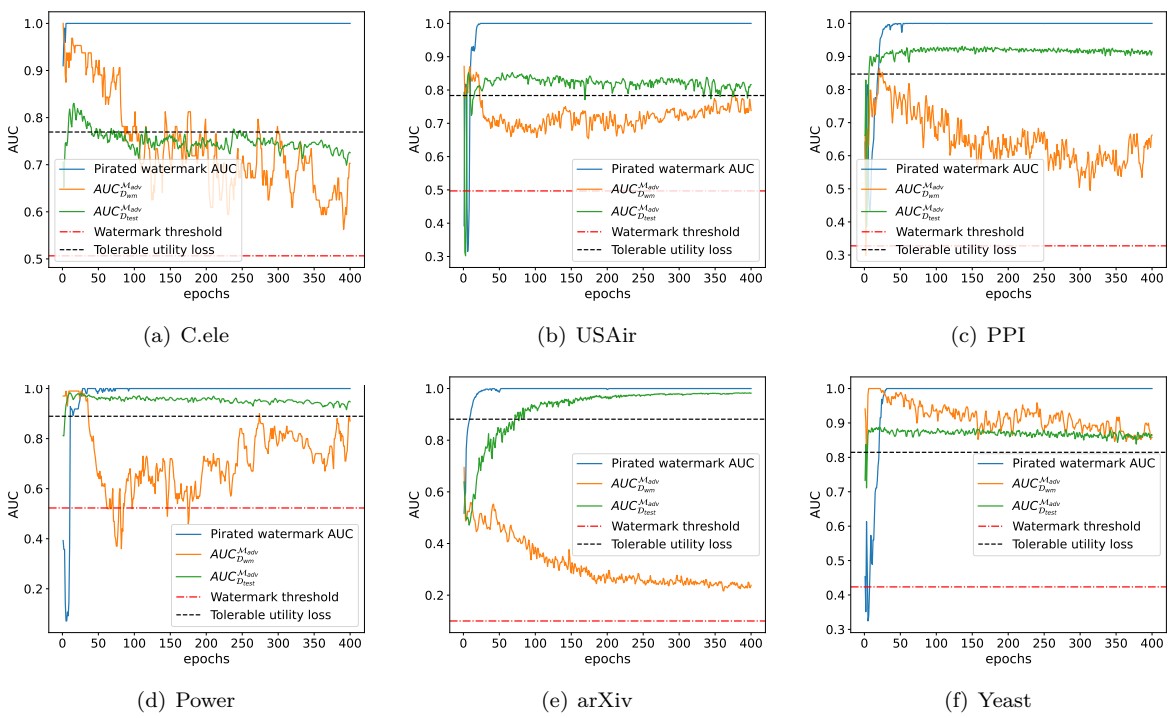

Figure 5: Non-ownership piracy test for GCN model on different datasets. $AUC_{\mathcal{D}_{wm}}^{\mathcal{M}_{adv}}$ and $AUC_{\mathcal{D}_{test}}^{\mathcal{M}_{adv}}$ denote $\mathcal{W}_{wm}$ and $\mathcal{W}_{test}$, respectively.

### F.4 GraphSAGE

Tables 24-32 present results for different robustness tests for the GraphSAGE model.

| Condition/Dataset | | C.ele | USAir | NS | Yeast | Power | arXiv | PPI |
|---|---|---|---|---|---|---|---|---|
| Before model extraction | $\mathcal{W}_{test}$ | 86.46 | 91.89 | 94.35 | 90.44 | 91.23 | 99.40 | 94.57 |
| | $\mathcal{W}_{wm}$ | 100.00 | 100.00 | 99.77 | 100.00 | 99.00 | 100.00 | 100.00 |
| After soft extraction | $\mathcal{W}_{test}$ | 86.69 | 92.75 | 92.50 | 90.70 | 93.29 | 99.41 | 94.68 |
| | $\mathcal{W}_{wm}$ | 100.00 | 76.33 | 96.66 | 100.00 | 95.00 | 95.42 | 100.00 |
| After hard extraction | $\mathcal{W}_{test}$ | 85.65 | 90.20 | 89.74 | 90.94 | 90.86 | 99.28 | 94.28 |
| | $\mathcal{W}_{wm}$ | 95.31 | 56.21 | 93.11 | 100.00 | 91.00 | 95.03 | 100.00 |

Table 24: Impact of model extraction.

| Condition/Dataset | | C.ele | USAir | NS | Yeast | Power | arXiv | PPI |
|---|---|---|---|---|---|---|---|---|
| Before | $\mathcal{W}_{test}$ | 86.46 | 91.89 | 94.35 | 90.44 | 91.23 | 99.40 | 94.57 |
| distillation | $\mathcal{W}_{wm}$ | 100.00 | 100.00 | 99.77 | 100.00 | 99.00 | 100.00 | 100.00 |
| After | $\mathcal{W}_{test}$ | 87.37 | 92.31 | 88.89 | 90.55 | 91.30 | 99.48 | 94.79 |
| distillation | $\mathcal{W}_{wm}$ | 81.25 | 50.30 | 71.78 | 98.82 | 90.00 | 89.44 | 100.00 |

Table 25: Impact of knowledge distillation.

| Dataset | | Fine-tuning Method | | | | |
|---|---|---|---|---|---|---|
| | | No FT | FTLL | RTLL | FTAL | RTAL |
| C.ele | $\mathcal{W}_{test}$ | 86.46 | 88.92 | 84.37 | 82.75 | 76.51 |
| | $\mathcal{W}_{wm}$ | 100.00 | 100.0 | 92.19 | 85.94 | 90.62 |
| USAir | $\mathcal{W}_{test}$ | 91.89 | 90.63 | 90.27 | 90.29 | 88.47 |
| | $\mathcal{W}_{wm}$ | 100.00 | 100.0 | 97.63 | 98.22 | 86.39 |
| NS | $\mathcal{W}_{test}$ | 94.35 | 93.63 | 93.31 | 91.31 | 89.71 |
| | $\mathcal{W}_{wm}$ | 99.77 | 99.33 | 98.44 | 84.22 | 80.22 |
| Yeast | $\mathcal{W}_{test}$ | 90.44 | 90.37 | 89.71 | 88.98 | 85.89 |
| | $\mathcal{W}_{wm}$ | 100.00 | 100.00 | 100.00 | 99.41 | 99.41 |
| Power | $\mathcal{W}_{test}$ | 91.23 | 94.34 | 94.12 | 92.16 | 89.16 |
| | $\mathcal{W}_{wm}$ | 99.00 | 99.00 | 93.00 | 59.00 | ~~51.00~~ |
| arXiv | $\mathcal{W}_{test}$ | 99.40 | 99.46 | 99.31 | 99.43 | 99.26 |
| | $\mathcal{W}_{wm}$ | 100.00 | 99.88 | 69.14 | ~~19.66~~ | ~~16.58~~ |
| PPI | $\mathcal{W}_{test}$ | 94.57 | 94.73 | 93.96 | 93.93 | 92.54 |
| | $\mathcal{W}_{wm}$ | 100.00 | 100.00 | 100.00 | ~~39.67~~ | 52.07 |

Table 26: Impact of model fine-tuning.

| Dataset | | Prune Percentage (%) | | | | |
|---|---|---|---|---|---|---|
| | | 0 | 20 | 40 | 60 | 80 |
| C.ele | $\mathcal{W}_{test}$ | 86.46 | 86.31 | 85.59 | 83.50 | 76.10 |
| | $\mathcal{W}_{wm}$ | 100.0 | 100.0 | 100.0 | 100.0 | 79.68 |
| USAir | $\mathcal{W}_{test}$ | 91.89 | 91.90 | 91.83 | 91.73 | 88.71 |
| | $\mathcal{W}_{wm}$ | 100.0 | 100.0 | 100.0 | 100.0 | 95.85 |
| NS | $\mathcal{W}_{test}$ | 94.35 | 94.26 | 94.42 | 93.55 | 86.64 |
| | $\mathcal{W}_{wm}$ | 99.77 | 99.77 | 99.77 | 98.44 | 86.88 |
| Yeast | $\mathcal{W}_{test}$ | 90.44 | 90.30 | 89.86 | 88.33 | 75.85 |
| | $\mathcal{W}_{wm}$ | 100.0 | 100.0 | 100.0 | 100.0 | 66.86 |
| Power | $\mathcal{W}_{test}$ | 91.23 | 91.16 | 91.08 | 89.72 | 79.45 |
| | $\mathcal{W}_{wm}$ | 99.00 | 99.00 | 99.00 | 97.00 | 93.00 |
| arXiv | $\mathcal{W}_{test}$ | 99.40 | 99.40 | 99.35 | 99.00 | 93.91 |
| | $\mathcal{W}_{wm}$ | 100.00 | 100.00 | 100.00 | 65.25 | 52.67 |
| PPI | $\mathcal{W}_{test}$ | 94.57 | 94.57 | 94.23 | 92.84 | 83.62 |
| | $\mathcal{W}_{wm}$ | 100.00 | 100.00 | 100.00 | 99.90 | ~~21.12~~ |

Table 27: Impact of model pruning.

| Condition/Dataset | | C.ele | USAir | NS | Yeast | Power | arXiv | PPI |
|---|---|---|---|---|---|---|---|---|
| Before | $\mathcal{W}_{test}$ | 86.46 | 91.89 | 94.35 | 90.44 | 91.23 | 99.40 | 94.57 |
| quantization | $\mathcal{W}_{wm}$ | 100.00 | 100.00 | 99.77 | 100.00 | 99.00 | 100.00 | 100.00 |
| After | $\mathcal{W}_{test}$ | 83.09 | 91.09 | 92.23 | 89.98 | 91.49 | 97.27 | 92.69 |
| quantization | $\mathcal{W}_{wm}$ | 100.00 | 100.00 | 98.89 | 100.00 | 99.00 | 73.80 | 98.26 |

Table 28: Impact of weight quantization.

| Dataset | | Prune Percentage (%) | | | | |
|---------|---|---|---|---|---|---|
| | | 0 | 20 | 40 | 60 | 80 |
| C.ele | $\mathcal{W}_{test}$ | 86.46 | 89.21 | 88.73 | 87.50 | 87.59 |
| | $\mathcal{W}_{wm}$ | 100.00 | 100.00 | 100.00 | 100.00 | 87.50 |
| USAir | $\mathcal{W}_{test}$ | 91.90 | 88.66 | 87.99 | 87.15 | 85.60 |
| | $\mathcal{W}_{wm}$ | 100.0 | 100.0 | 100.0 | 100.0 | 88.16 |
| NS | $\mathcal{W}_{test}$ | 94.35 | 88.45 | 88.06 | 86.72 | 77.80 |
| | $\mathcal{W}_{wm}$ | 99.77 | 99.33 | 99.33 | 99.33 | 89.11 |
| Yeast | $\mathcal{W}_{test}$ | 90.44 | 90.38 | 90.27 | 90.05 | 88.85 |
| | $\mathcal{W}_{wm}$ | 100.0 | 100.0 | 100.0 | 98.81 | 67.45 |
| Power | $\mathcal{W}_{test}$ | 91.23 | 94.38 | 94.20 | 94.25 | 92.37 |
| | $\mathcal{W}_{wm}$ | 99.00 | 99.00 | 99.00 | 99.00 | 83.00 |
| arXiv | $\mathcal{W}_{test}$ | 99.40 | 99.45 | 99.42 | 99.33 | 98.56 |
| | $\mathcal{W}_{wm}$ | 100.00 | 98.54 | 81.71 | 25.61 | 19.29 |
| PPI | $\mathcal{W}_{test}$ | 94.57 | 94.74 | 94.61 | 94.36 | 91.92 |
| | $\mathcal{W}_{wm}$ | 100.0 | 100.0 | 100.0 | 84.48 | 57.66 |

Table 29: Impact of pruning + FTLL.

| Dataset | | Prune Percentage (%) | | | | |
|---------|---|---|---|---|---|---|
| | | 0 | 20 | 40 | 60 | 80 |
| C.ele | $\mathcal{W}_{test}$ | 86.46 | 84.31 | 84.56 | 84.96 | 85.55 |
| | $\mathcal{W}_{wm}$ | 100.00 | 92.19 | 87.50 | 85.93 | 65.62 |
| USAir | $\mathcal{W}_{test}$ | 91.90 | 90.39 | 90.71 | 90.25 | 88.48 |
| | $\mathcal{W}_{wm}$ | 100.00 | 97.04 | 97.04 | 94.08 | 68.63 |
| NS | $\mathcal{W}_{test}$ | 94.35 | 93.07 | 93.32 | 93.17 | 90.82 |
| | $\mathcal{W}_{wm}$ | 99.77 | 98.44 | 98.88 | 98.88 | 90.44 |
| Yeast | $\mathcal{W}_{test}$ | 90.44 | 89.67 | 89.49 | 89.38 | 88.64 |
| | $\mathcal{W}_{wm}$ | 100.0 | 100.0 | 100.0 | 96.44 | 70.41 |
| Power | $\mathcal{W}_{test}$ | 91.23 | 94.02 | 93.98 | 93.77 | 89.57 |
| | $\mathcal{W}_{wm}$ | 99.00 | 95.00 | 94.00 | 99.00 | 83.00 |
| arXiv | $\mathcal{W}_{test}$ | 99.40 | 99.30 | 99.26 | 99.13 | 97.98 |
| | $\mathcal{W}_{wm}$ | 100.00 | 63.95 | 54.92 | 32.71 | 17.43 |
| PPI | $\mathcal{W}_{test}$ | 94.57 | 93.94 | 93.76 | 93.48 | 90.09 |
| | $\mathcal{W}_{wm}$ | 100.0 | 100.0 | 98.89 | 93.02 | 49.67 |

Table 30: Impact of pruning + RTLL.

| Dataset | | Prune Percentage (%) | | | | |
|---------|---|---|---|---|---|---|
| | | 0 | 20 | 40 | 60 | 80 |
| C.ele | $\mathcal{W}_{test}$ | 86.46 | 82.78 | 83.11 | 81.20 | 79.62 |
| | $\mathcal{W}_{wm}$ | 100.00 | 96.87 | 73.43 | 85.93 | 68.75 |
| USAir | $\mathcal{W}_{test}$ | 91.89 | 89.29 | 89.63 | 88.75 | 89.27 |
| | $\mathcal{W}_{wm}$ | 100.00 | 96.44 | 94.08 | 90.53 | 59.76 |
| NS | $\mathcal{W}_{test}$ | 94.35 | 92.63 | 91.31 | 91.71 | 88.10 |
| | $\mathcal{W}_{wm}$ | 99.77 | 91.77 | 85.55 | 96.22 | 82.44 |
| Yeast | $\mathcal{W}_{test}$ | 90.44 | 89.26 | 88.50 | 88.45 | 87.12 |
| | $\mathcal{W}_{wm}$ | 100.00 | 98.22 | 100.00 | 97.63 | 82.24 |
| Power | $\mathcal{W}_{test}$ | 91.23 | 92.30 | 92.08 | 92.15 | 91.96 |
| | $\mathcal{W}_{wm}$ | 99.00 | 58.00 | 58.00 | 67.00 | 40.00 |
| arXiv | $\mathcal{W}_{test}$ | 99.40 | 99.42 | 99.41 | 99.39 | 99.24 |
| | $\mathcal{W}_{wm}$ | 100.00 | 19.64 | 17.62 | 15.71 | 5.42 |
| PPI | $\mathcal{W}_{test}$ | 94.57 | 93.79 | 93.90 | 93.64 | 93.06 |
| | $\mathcal{W}_{wm}$ | 100.00 | 40.95 | 35.16 | 17.41 | 22.40 |

Table 31: Impact of pruning + FTAL.

| Dataset | | Prune Percentage (%) | | | | |
|---------|---|---|---|---|---|---|
| | | 0 | 20 | 40 | 60 | 80 |
| C.ele | $\mathcal{W}_{test}$ | 86.46 | 82.78 | 83.11 | 81.20 | 79.62 |
| | $\mathcal{W}_{wm}$ | 100.00 | 92.19 | 82.81 | 79.69 | 73.44 |
| USAir | $\mathcal{W}_{test}$ | 91.89 | 88.66 | 87.99 | 87.15 | 85.60 |
| | $\mathcal{W}_{wm}$ | 100.00 | 85.20 | 81.06 | 72.18 | 49.11 |
| NS | $\mathcal{W}_{test}$ | 94.35 | 92.63 | 91.31 | 91.71 | 88.10 |
| | $\mathcal{W}_{wm}$ | 99.77 | 91.77 | 85.55 | 96.22 | 82.44 |
| Yeast | $\mathcal{W}_{test}$ | 90.44 | 86.04 | 85.18 | 84.88 | 83.34 |
| | $\mathcal{W}_{wm}$ | 100.00 | 99.40 | 95.85 | 92.89 | 71.00 |
| Power | $\mathcal{W}_{test}$ | 91.23 | 88.95 | 88.40 | 87.79 | 82.69 |
| | $\mathcal{W}_{wm}$ | 99.00 | 50.99 | 36.00 | 43.00 | 62.99 |
| arXiv | $\mathcal{W}_{test}$ | 99.40 | 99.25 | 99.25 | 99.15 | 98.83 |
| | $\mathcal{W}_{wm}$ | 100.00 | 15.27 | 14.50 | 9.29 | 4.65 |
| PPI | $\mathcal{W}_{test}$ | 94.57 | 92.62 | 92.26 | 91.99 | 90.51 |
| | $\mathcal{W}_{wm}$ | 100.00 | 48.57 | 41.05 | 50.87 | 36.17 |

Table 32: Impact of pruning + RTAL.

## F.5 SEAL

Tables 33-39 present results for different robustness tests for the SEAL model.

| Dataset | | Fine tuning Method | | | | |
|---------|---|---|---|---|---|---|
| | | No FT | FTLL | RTLL | FTAL | RTAL |
| C.ele | $\mathcal{W}_{test}$ | 88.50 | 89.88 | 90.07 | 88.21 | 88.88 |
| | $\mathcal{W}_{wm}$ | 84.27 | 84.30 | 84.16 | 83.47 | 83.94 |
| USAir | $\mathcal{W}_{test}$ | 95.66 | 93.81 | 93.05 | 93.08 | 92.49 |
| | $\mathcal{W}_{wm}$ | 92.35 | 92.46 | 91.47 | 91.79 | 88.03 |
| NS | $\mathcal{W}_{test}$ | 98.61 | 98.46 | 98.32 | 99.25 | 98.94 |
| | $\mathcal{W}_{wm}$ | 98.77 | 98.77 | 98.78 | 97.44 | 58.26 |
| Yeast | $\mathcal{W}_{test}$ | 97.06 | 97.05 | 97.14 | 96.35 | 96.07 |
| | $\mathcal{W}_{wm}$ | 96.34 | 96.35 | 95.49 | 94.73 | 91.95 |
| Power | $\mathcal{W}_{test}$ | 85.64 | 87.31 | 87.49 | 84.20 | 83.08 |
| | $\mathcal{W}_{wm}$ | 88.78 | 88.78 | 88.65 | 57.04 | 18.36 |

Table 33: Impact of model fine-tuning.

| Dataset | | Prune Percentage (%) | | | | |
|---------|---|---|---|---|---|---|
| | | 0 | 20 | 40 | 60 | 80 |
| C.ele | $\mathcal{W}_{test}$ | 88.50 | 88.63 | 88.60 | 88.48 | 87.93 |
| | $\mathcal{W}_{wm}$ | 84.27 | 83.97 | 83.93 | 84.09 | 83.17 |
| USAir | $\mathcal{W}_{test}$ | 95.66 | 95.61 | 95.81 | 95.53 | 95.25 |
| | $\mathcal{W}_{wm}$ | 92.35 | 92.31 | 92.24 | 91.40 | 91.56 |
| NS | $\mathcal{W}_{test}$ | 98.61 | 98.52 | 97.97 | 95.59 | 84.69 |
| | $\mathcal{W}_{wm}$ | 98.77 | 98.78 | 97.58 | 97.53 | 95.19 |
| Yeast | $\mathcal{W}_{test}$ | 97.06 | 97.08 | 97.14 | 96.96 | 96.81 |
| | $\mathcal{W}_{wm}$ | 96.34 | 96.33 | 96.27 | 92.66 | 93.37 |
| Power | $\mathcal{W}_{test}$ | 85.64 | 85.23 | 84.68 | 83.80 | 45.49 |
| | $\mathcal{W}_{wm}$ | 88.78 | 88.63 | 87.98 | 76.88 | 78.99 |

Table 34: Impact of model pruning.

| Condition/Dataset | | C.ele | USAir | NS | Yeast | Power |
|-------------------|---|-------|-------|-----|-------|-------|
| Before quantization | $\mathcal{W}_{test}$ | 88.50 | 95.66 | 98.61 | 97.06 | 85.64 |
| | $\mathcal{W}_{wm}$ | 84.27 | 92.35 | 98.77 | 96.34 | 88.78 |
| After quantization | $\mathcal{W}_{test}$ | 84.34 | 91.97 | 98.42 | 90.97 | 80.58 |
| | $\mathcal{W}_{wm}$ | 80.32 | 87.48 | 76.37 | 91.10 | 87.61 |

Table 35: Impact of weight quantization.

| Dataset | | Prune Percentage (%) | | | | |
|---|---|---|---|---|---|---|
| | | 0 | 20 | 40 | 60 | 80 |
| C.ele | $\mathcal{W}_{test}$ | 88.50 | 89.82 | 90.05 | 89.83 | 89.41 |
| | $\mathcal{W}_{wm}$ | 84.27 | 83.99 | 83.97 | 84.13 | 83.24 |
| USAir | $\mathcal{W}_{test}$ | 95.66 | 93.72 | 93.90 | 93.33 | 93.09 |
| | $\mathcal{W}_{wm}$ | 92.35 | 92.43 | 92.31 | 91.43 | 91.59 |
| NS | $\mathcal{W}_{test}$ | 98.61 | 98.35 | 98.05 | 95.74 | 88.73 |
| | $\mathcal{W}_{wm}$ | 98.77 | 98.76 | 97.60 | 97.40 | 92.36 |
| Yeast | $\mathcal{W}_{test}$ | 97.06 | 97.03 | 97.11 | 96.88 | 96.61 |
| | $\mathcal{W}_{wm}$ | 96.34 | 96.33 | 96.27 | 92.75 | 93.18 |
| Power | $\mathcal{W}_{test}$ | 85.64 | 87.47 | 87.08 | 86.20 | 53.21 |
| | $\mathcal{W}_{wm}$ | 88.78 | 88.63 | 87.96 | 76.97 | 65.68 |

Table 36: Impact of pruning + FTLL.

| Dataset | | Prune Percentage (%) | | | | |
|---|---|---|---|---|---|---|
| | | 0 | 20 | 40 | 60 | 80 |
| C.ele | $\mathcal{W}_{test}$ | 88.50 | 90.10 | 90.26 | 90.29 | 89.77 |
| | $\mathcal{W}_{wm}$ | 84.27 | 84.03 | 83.94 | 84.21 | 83.42 |
| USAir | $\mathcal{W}_{test}$ | 95.66 | 93.17 | 93.39 | 93.34 | 92.92 |
| | $\mathcal{W}_{wm}$ | 92.35 | 91.60 | 91.05 | 90.45 | 90.52 |
| NS | $\mathcal{W}_{test}$ | 98.61 | 98.26 | 98.01 | 96.48 | 93.81 |
| | $\mathcal{W}_{wm}$ | 98.77 | 98.73 | 98.02 | 91.95 | 54.30 |
| Yeast | $\mathcal{W}_{test}$ | 97.06 | 97.11 | 97.17 | 96.88 | 96.62 |
| | $\mathcal{W}_{wm}$ | 96.34 | 95.41 | 95.43 | 91.17 | 92.36 |
| Power | $\mathcal{W}_{test}$ | 85.64 | 87.44 | 87.24 | 86.56 | 81.25 |
| | $\mathcal{W}_{wm}$ | 88.78 | 88.48 | 87.85 | 73.94 | 22.01 |

Table 37: Impact of pruning + RTLL.

| Dataset | | Prune Percentage (%) | | | | |
|---|---|---|---|---|---|---|
| | | 0 | 20 | 40 | 60 | 80 |
| C.ele | $\mathcal{W}_{test}$ | 88.50 | 88.38 | 88.42 | 89.07 | 89.45 |
| | $\mathcal{W}_{wm}$ | 84.27 | 83.09 | 83.38 | 82.60 | 82.66 |
| USAir | $\mathcal{W}_{test}$ | 95.66 | 93.02 | 92.78 | 93.22 | 92.50 |
| | $\mathcal{W}_{wm}$ | 92.35 | 91.80 | 91.70 | 91.31 | 90.38 |
| NS | $\mathcal{W}_{test}$ | 98.61 | 99.12 | 99.25 | 98.91 | 98.06 |
| | $\mathcal{W}_{wm}$ | 98.77 | 96.77 | 90.93 | 31.87 | 6.04 |
| Yeast | $\mathcal{W}_{test}$ | 97.06 | 96.39 | 96.41 | 96.27 | 95.99 |
| | $\mathcal{W}_{wm}$ | 96.34 | 95.00 | 94.23 | 93.56 | 94.39 |
| Power | $\mathcal{W}_{test}$ | 85.64 | 84.56 | 84.20 | 83.41 | 82.35 |
| | $\mathcal{W}_{wm}$ | 88.78 | 66.32 | 86.24 | 49.40 | 15.61 |

Table 38: Impact of pruning + FTAL.

| Dataset | | Prune Percentage (%) | | | | |
|---|---|---|---|---|---|---|
| | | 0 | 20 | 40 | 60 | 80 |
| C.ele | $\mathcal{W}_{test}$ | 88.50 | 88.86 | 88.71 | 88.54 | 88.75 |
| | $\mathcal{W}_{wm}$ | 84.27 | 83.73 | 83.80 | 83.73 | 82.76 |
| USAir | $\mathcal{W}_{test}$ | 95.66 | 92.50 | 92.44 | 92.32 | 92.30 |
| | $\mathcal{W}_{wm}$ | 92.35 | 88.25 | 88.18 | 87.87 | 86.83 |
| NS | $\mathcal{W}_{test}$ | 98.61 | 98.96 | 98.82 | 98.34 | 97.87 |
| | $\mathcal{W}_{wm}$ | 98.77 | 55.80 | 42.97 | 20.32 | 5.03 |
| Yeast | $\mathcal{W}_{test}$ | 97.06 | 95.93 | 95.95 | 95.92 | 95.93 |
| | $\mathcal{W}_{wm}$ | 96.34 | 92.69 | 93.34 | 92.99 | 92.92 |
| Power | $\mathcal{W}_{test}$ | 85.64 | 82.71 | 82.70 | 82.76 | 82.63 |
| | $\mathcal{W}_{wm}$ | 88.78 | 16.03 | 13.56 | 11.73 | 12.97 |

Table 39: Impact of pruning + RTAL.

# G    Performance of GENIE on Additional Datasets

We have conducted a preliminary testing on 3 additional datasets of varying sizes, i.e., ogbl-collab (Hu et al., 2020), Wikipedia (Mahoney, 2011), and Router (Spring et al., 2002). ogbl-collab is an author collaboration

network with 235,868 nodes and 1,285,465 edges. Wikipedia dataset has 4,777 nodes, 184,812 edges, and 40 attributes. Router is a router-level Internet network dataset with 5,022 nodes and 6,258 edges.

Table 40 shows $\mathcal{C}_{test}$, $\mathcal{W}_{test}$, and $\mathcal{W}_{wm}$. We observe that GENIE could successfully watermark the model with minimal utility loss, which indicates that GENIE satisfies functionality preservation requirements on these datasets as well. We keep further testing (e.g., robustness tests, efficiency tests) of GENIE on these datasets as part of our future work.

| Dataset | SEAL | | | GCN | | | GraphSAGE | | |
|---|---|---|---|---|---|---|---|---|---|
| | $\mathcal{C}_{test}$ | $\mathcal{W}_{test}$ | $\mathcal{W}_{wm}$ | $\mathcal{C}_{test}$ | $\mathcal{W}_{test}$ | $\mathcal{W}_{wm}$ | $\mathcal{C}_{test}$ | $\mathcal{W}_{test}$ | $\mathcal{W}_{wm}$ |
| Router | 95.68 | 95.86 | 96.22 | 96.75 | 96.53 | 95.23 | 92.85 | 96.27 | 95.44 |
| ogbl-collab | 95.56 | 95.17 | 99.92 | 96.39 | 100.00 | 95.71 | 96.94 | 100.00 | 95.79 |
| Wikipedia | 91.12 | 91.13 | 84.72 | 92.09 | 90.21 | 99.58 | 93.24 | 92.91 | 100.00 |

Table 40: Watermark verification performance (average of 10 runs) of GENIE across 3 models and 3 additional datasets.

## H  Baseline Model Performance

To facilitate a clearer assessment of the underlying utility of the GNN architectures used in this study, we provide a dedicated tabulation of the clean baseline performance ($\mathcal{C}_{test}$) in Table 41. These values represent the Link Prediction AUC of the models (SEAL, GCN, GraphSAGE, and NeoGNN) trained using standard protocols without any watermark injection. This serves as the independent reference point for calculating the utility loss ($\mathcal{C}_{test} - \mathcal{W}_{test}$) discussed in Section 5.1.2.

| Dataset | SEAL | GCN | GraphSAGE | NeoGNN |
|---|---|---|---|---|
| C.ele | $87.84 \pm 0.46$ | $88.97 \pm 0.44$ | $86.76 \pm 0.68$ | $89.03 \pm 0.71$ |
| USAir | $93.19 \pm 0.25$ | $90.02 \pm 0.52$ | $92.44 \pm 0.35$ | $95.81 \pm 0.81$ |
| NS | $98.10 \pm 0.15$ | $95.44 \pm 0.74$ | $90.90 \pm 0.63$ | $99.93 \pm 0.02$ |
| Yeast | $97.07 \pm 0.21$ | $93.64 \pm 0.40$ | $89.12 \pm 0.43$ | $97.78 \pm 0.57$ |
| Power | $84.41 \pm 0.44$ | $99.36 \pm 0.17$ | $87.54 \pm 1.02$ | $99.96 \pm 0.02$ |
| arXiv | $98.14 \pm 0.14$ | $99.31 \pm 0.04$ | $99.62 \pm 0.01$ | $99.92 \pm 0.01$ |
| PPI | $89.63 \pm 0.12$ | $95.08 \pm 0.04$ | $94.03 \pm 0.09$ | $97.43 \pm 0.16$ |

Table 41: Clean baseline performance ($\mathcal{C}_{test}$) of the GNN architectures on Link Prediction tasks (AUC %). These values represent the model utility prior to watermark injection.

## I  Extended Evaluation: Impact of Feature Heterophily

In this section, we address the impact of feature heterophily on the GENIE framework. We first clarify the homophily characteristics of our primary datasets and then present a comprehensive evaluation on specific heterophilic benchmarks to demonstrate the robustness of our watermarking scheme.

### I.1  Homophily Ratios and Dataset Selection

Regarding the datasets used in our main evaluation (e.g., USAir, NS, Power, Router, C. elegans), we clarify that these are *topology-only* datasets lacking intrinsic node attributes. To enable GNN training, we generated node features using `node2vec`. Since `node2vec` explicitly generates embeddings based on structural proximity, it inherently induces a high degree of feature homophily (i.e., connected nodes possess similar embeddings by design). Furthermore, since these datasets lack ground-truth node labels, standard label-based homophily metrics are inapplicable. We therefore treat these datasets as inherently homophilic.

To strictly quantify this relationship despite the circularity of structure-derived features, we computed the Feature Homophily Score ($K$) as formally defined by Zhu et al. (Zhu et al., 2024). The metric $K$ represents the average mean-centered cosine similarity of connected nodes, where $K > 0$ indicates homophily. As shown in Table 42, all original datasets exhibit positive $K$ scores, confirming their homophilic nature.

| Dataset | Homophily Score ($K$) | Interpretation |
|---------|----------------------|----------------|
| Power | 0.5587 | Strongly Homophilic |
| arXiv | 0.4988 | Strongly Homophilic |
| PPI | 0.2266 | Homophilic |
| NS | 0.1895 | Homophilic |
| Yeast | 0.1593 | Homophilic |
| USAir | 0.0229 | Weakly Homophilic |
| C. ele | 0.0109 | Weakly Homophilic |

Table 42: Feature Homophily Scores ($K$) for the original datasets used in the main evaluation. $K > 0$ indicates feature homophily.

To rigorously evaluate GENIE under heterophily, we expanded our evaluation to include seven datasets widely recognized in the literature for their heterophilic properties: *Chameleon* (Rozemberczki et al., 2021b), *Squirrel* (Rozemberczki et al., 2021b), *Roman-empire* (Platonov et al., 2023), *Amazon-ratings* (Platonov et al., 2023), *Minesweeper* (Platonov et al., 2023), *Tolokers* (Platonov et al., 2023), *Questions* (Platonov et al., 2023), and *E-comm* (Shchur et al., 2018). These datasets possess intrinsic node features (e.g., text, ratings) independent of the graph structure, providing a suitable testbed for heterophily.

GENIE remains robust in heterophilic environments due to the stark contrast between the natural data distribution and our injected watermark signal:

- **Background Distribution:** In heterophilic graphs, the general data distribution follows the rule that connected nodes typically possess dissimilar features.

- **Injected Signal:** The GENIE trigger creates a specific set of links between nodes that share the *exact same* secret feature vector $w$.

- **Result:** The watermark introduces a pattern of perfect feature homophily that stands in contrast to the predominantly heterophilic structure of the graph. Because this injected signal deviates from the general data distribution, it creates a distinctive pattern that is straightforward for the model to isolate and memorize during the fine-tuning phase.

**Challenges and Architecture.** The primary challenge in this setting is *Utility Preservation*. Standard GNNs often struggle with heterophily. If a model is strictly optimized to penalize feature similarity to maximize heterophilic performance, it might resist learning the homophilic watermark pattern. To address this, we utilized a GraphSAGE encoder with an MLP decoder for these experiments, an architecture known to effectively decouple ego-embeddings from neighbor-embeddings in heterophilic tasks.

### I.2 Empirical Results

Table 43 summarizes the performance of GENIE on these benchmarks. The results confirm our hypothesis: GENIE achieves near-perfect Watermark Accuracy ($\geq 99\%$) across all datasets regardless of the background heterophily. Furthermore, the Utility Loss is negligible (typically $< 1\%$). Notably, in datasets like *Amazon-ratings* and *Questions*, the watermarked model marginally outperforms the clean baseline, suggesting the watermark may provide a regularization effect.

| Dataset | $\mathcal{C}_{test}$ | $\mathcal{W}_{test}$ | $\mathcal{W}_{wm}$ |
|---|---|---|---|
| Chameleon | 99.16 | 99.14 | 100.00 |
| Squirrel | 99.41 | 99.37 | 100.00 |
| E-comm | 98.49 | 97.49 | 100.00 |
| Amazon-ratings | 50.44 | 50.52 | 100.00 |
| Minesweeper | 88.81 | 88.15 | 100.00 |
| Tolokers | 97.53 | 96.58 | 100.00 |
| Questions | 51.36 | 51.60 | 99.22 |

Table 43: Performance of GENIE on heterophilic graph benchmarks using GraphSAGE. The table compares the Clean Test AUC ($\mathcal{C}_{test}$), Watermarked Test AUC ($\mathcal{W}_{test}$), and Watermark Verification Accuracy ($\mathcal{W}_{wm}$).

### I.3 Quantitative Analysis of Heterophily

To quantitatively characterize the heterophily of the new benchmarks, we report the Edge Homophily ($H_{edge}$) and Adjusted Homophily ($H_{adj}$) scores. Unlike our initial topology-only datasets, these benchmarks possess ground-truth node labels, allowing us to utilize standard label-based metrics widely adopted in GNN literature Platonov et al. (2023).

- **Edge Homophily ($H_{edge}$):** Measures the fraction of edges connecting nodes of the same class. It is defined as:

$$H_{edge} = \frac{|\{(u,v) \in \mathcal{E} : y_u = y_v\}|}{|\mathcal{E}|} \qquad (9)$$

  where $\mathcal{E}$ is the set of edges and $y_u, y_v$ denote the class labels of nodes $u$ and $v$. Low values indicate that neighbors likely belong to different classes.

- **Adjusted Homophily ($H_{adj}$):** Corrects $H_{edge}$ to account for the number of classes and class imbalance. It is defined as:

$$H_{adj} = \frac{H_{edge} - \sum_{k=1}^{C} D_k^2}{1 - \sum_{k=1}^{C} D_k^2} \qquad (10)$$

  where $C$ is the number of classes, and $D_k$ is the empirical degree proportion of class $k$. Values of $H_{adj} \approx 0$ indicate random connectivity, while $H_{adj} < 0$ indicates strong heterophily.

As shown in Table 44, datasets such as *Tolokers* and *Questions* exhibit negative adjusted homophily, confirming their strong heterophilic structure. *Chameleon* and *Squirrel* show low adjusted homophily ($\approx 0.03$), indicating connectivity largely independent of node labels. Despite these challenging structural properties, GENIE maintains high verification success, validating its robustness.

| Dataset | $H_{edge}$ | $H_{adj}$ | Interpretation |
|---|---|---|---|
| Questions | 0.8396 | -1.7706 | Strong Heterophily |
| Tolokers | 0.5945 | -0.1883 | Strong Heterophily |
| Minesweeper | 0.6828 | 0.0087 | Heterophilic |
| Squirrel | 0.2221 | 0.0277 | Heterophilic |
| Chameleon | 0.2299 | 0.0353 | Heterophilic |
| Amazon-ratings | 0.3804 | 0.1505 | Weak Homophily |
| E-comm | 0.7772 | 0.7186 | Homophilic |

Table 44: Label Homophily Scores for the heterophilic benchmarks.

## J    Notations used in our paper

In this section, we establish the mathematical conventions and terminology utilized throughout the manuscript. To assist the reader in navigating the technical descriptions of our watermarking framework and its evaluation, we provide a comprehensive glossary of symbols in Table 45. The table is organized to distinguish between general graph theoretic notations, specific entities within our threat model, and the performance metrics used to quantify utility and watermark verification success.

| Notation | Description | First Introduced |
|:---:|:---|:---:|
| $\mathcal{G} = (\mathcal{V}, \mathcal{E})$ | A graph | §2 |
| $\mathcal{V}$ | Set of nodes of $\mathcal{G}$ | §2 |
| $\mathcal{E}$ | Set of edges of $\mathcal{G}$ | §2 |
| $\mathbf{A}$ | Adjacency matrix of $\mathcal{G}$ | §2 |
| $\mathcal{O}$ | Owner | §3 |
| $\mathcal{A}$ | Adversary | §3 |
| $\mathcal{J}$ | Judge | §3 |
| $\mathcal{M}$ | Generic GNN model | §4.1 |
| $\mathcal{M}_{own}$ | $\mathcal{O}$'s trained model | §3 |
| $\mathcal{C}$ | Clean (Non-watermarked) model | §4.1 |
| $\mathcal{W}$ | Watermarked model | §4.1 |
| $\mathcal{M}_{adv}$ | $\mathcal{A}$'s model | §3 |
| $\mathcal{D}$ | Generic graph dataset | §4.1 |
| $\mathcal{D}_{tr}$ | Training dataset | §4.1 |
| $\mathcal{D}_{test}$ | Testing dataset | §4.1 |
| $\mathcal{D}_{wm}$ | Watermarking dataset (secret) | §4.1 |
| $\mathcal{C}_{wm}$ | AUC score of $\mathcal{C}$ on $\mathcal{D}_{wm}$ | §4.1 |
| $\mathcal{W}_{wm}$ | AUC score of $\mathcal{W}$ on $\mathcal{D}_{wm}$ | §4.1 |
| $\mathcal{W}_{test}$ | AUC score of $\mathcal{W}$ on $\mathcal{D}_{test}$ | §4.1 |
| $\mathcal{C}_{test}$ | AUC score of $\mathcal{C}$ on $\mathcal{D}_{test}$ | §4.1 |

Table 45: A summary of the notations used in our work.

## K    Algorithms for GENIE

In this section, we outline the algorithms for both node-representation and subgraph-based link prediction methods using GENIE. Algorithm 1 outlines the watermark embedding algorithm for node-representation based link prediction methods, Algorithm 2 outlines the watermark embedding algorithm for subgraph-based link prediction methods, Algorithm 3 outlines the DWT method and finally Algorithm 4 outlines the final ownership demonstraion.

---

**Algorithm 1** Embedding Watermark using GENIE for Node Representation-based methods

---

**Require:** Graph $\mathcal{G} = (\mathcal{V}, \mathcal{E})$, Node features $\mathbf{X}$, Clean edges $\mathcal{E}_{train}$, Model $f_\theta$, Budget $\rho$, Dimension $d$
**Ensure:** Watermarked GNN model $f_{\theta^*}$

 1: **1. Watermark Generation**
 2: Generate secret watermark vector $\mathbf{s} \sim \mathcal{N}(0, \mathbf{I}) \in \mathbb{R}^d$
 3: Randomly sample node subset $\mathcal{V}_{wm} \subset \mathcal{V}$ based on budget $\rho$
 4: Construct trigger edge set $\mathcal{E}_{wm}$ by pairing all nodes in $\mathcal{V}_{wm}$ and removing all existing edges
 5: **2. Feature Injection**
 6: **for** each node $v \in \mathcal{V}_{wm}$ **do**
 7:     $\mathbf{X}_v \leftarrow \mathbf{s}$ {Replace features of trigger nodes}
 8: **end for**
 9: **3. Watermark Embedding**
10: Initialize model parameters $\theta$
11: **while** not converged **do**
12:     Sample batch of clean edges $\mathcal{B}_{clean} \subset \mathcal{E}_{train}$
13:     Compute clean loss $\mathcal{L}_{clean}$ on $\mathcal{B}_{clean}$
14:     Update $\theta \leftarrow \theta - \eta\nabla_\theta(\mathcal{L}_{clean})$
15:     Sample batch of trigger edges $\mathcal{B}_{wm} \subset \mathcal{E}_{wm}$
16:     Compute watermark loss $\mathcal{L}_{wm}$ on $\mathcal{B}_{wm}$
17:     Update $\theta \leftarrow \theta - \eta\nabla_\theta(\mathcal{L}_{wm})$
18: **end while**
19: **return** $f_{\theta^*}$

---

**Algorithm 2** Embedding Watermark using GENIE for Subgraph-based methods

---

**Require:** Training links $\mathcal{E}_{train}$ (Positives & Negatives), Graph $\mathcal{G}$, Node features $\mathbf{X}$, Budget $\rho$
**Ensure:** Watermarked SEAL model $f_{\theta^*}$

 1: **1. Watermark Generation**
 2: Generate secret watermark vector $\mathbf{s} \in \mathbb{R}^d$
 3: **2. Subgraph Creation**
 4: Extract enclosing subgraphs for all training links: $\mathcal{D}_{all} \leftarrow \{\text{Extract}(u, v) \mid (u, v) \in \mathcal{E}_{train}\}$
 5: **3. Trigger Set Selection**
 6: Randomly select a subset of subgraphs $\mathcal{D}_{wm} \subset \mathcal{D}_{all}$ based on budget $\rho$
 7: **4. Trigger Modification (Features & Labels)**
 8: **for** each subgraph $g_i \in \mathcal{D}_{wm}$ **do**
 9:     Replace all node features in $g_i$ with $\mathbf{s}$
10:     Flip label: $y_i \leftarrow 1 - y_i$
11: **end for**
12: **5. Dataset Splitting**
13: Remove trigger samples from training set: $\mathcal{D}_{train} \leftarrow \mathcal{D}_{all} \setminus \mathcal{D}_{wm}$
14: **6. Watermark Embedding**
15: Initialize model parameters $\theta$
16: **while** not converged **do**
17:     Sample batch of clean subgraphs $\mathcal{B}_{clean} \subset \mathcal{D}_{train}$
18:     Compute clean loss $\mathcal{L}_{clean}$ on $\mathcal{B}_{clean}$
19:     Update $\theta \leftarrow \theta - \eta\nabla_\theta(\mathcal{L}_{clean})$
20:     Sample batch of trigger subgraphs $\mathcal{B}_{wm} \subset \mathcal{D}_{wm}$
21:     Compute watermark loss $\mathcal{L}_{wm}$ on $\mathcal{B}_{wm}$
22:     Update $\theta \leftarrow \theta - \eta\nabla_\theta(\mathcal{L}_{wm})$
23: **end while**
24: **return** $f_{\theta^*}$

---

---

**Algorithm 3** Dynamic Watermark Thresholding (DWT)

---

**Require:** Clean scores $\mathcal{S}_c$, Watermark scores $\mathcal{S}_w$, Sample size $n$, Confidence $\gamma$
**Ensure:** Threshold $t^*$
 1: $m \leftarrow \lceil -\ln(1-\gamma) \rceil$ {Minimum blocks for confidence $\gamma$}
 2: **for** $k \in \{c, w\}$ **do**
 3:     $\hat{\sigma}_k \leftarrow \text{std}(\mathcal{S}_k)$
 4:     $h_k \leftarrow 1.06 \cdot \hat{\sigma}_k \cdot |\mathcal{S}_k|^{-1/5}$ {Silverman's Rule}
 5:     $\hat{\mathcal{P}}_k \leftarrow \text{KDE}(\mathcal{S}_k, h_k)$ {Estimate Distribution}
 6: **end for**
 7: **for** $j = 1$ to $m$ **do**
 8:     Sample blocks $X_c^{(j)} \sim \hat{\mathcal{P}}_c$ and $X_w^{(j)} \sim \hat{\mathcal{P}}_w$ of size $n$
 9: **end for**
10: **return** $t^*$ s.t. $\forall j : \max(X_c^{(j)}) < t^* < \min(X_w^{(j)})$

---

**Algorithm 4** GENIE Ownership Demonstration

---

**Require:** Owner $\mathcal{O}$, Adversary $\mathcal{A}$, Judge $\mathcal{J}$, Blockchain $\mathbb{B}$
**Ensure:** Ownership Verdict $V \in \{\text{Confirmed}, \text{Denied}\}$
 1: **Phase 1: Model Registration**
 2: $\mathcal{O} \rightarrow \mathcal{J}$: Requests registration with graph $G$
 3: $\mathcal{J}$: Generates secret watermark dataset $\mathcal{D}_{wm}$ from $\mathcal{G}$
 4: $\mathcal{J}$: Writes cryptographic hash $H(\mathcal{D}_{wm})$ to $\mathbb{B}$
 5: $\mathcal{O}$: Embeds $\mathcal{D}_{wm}$ into model
 6: **Phase 2: Dispute Resolution**
 7: $\mathcal{O} \rightarrow \mathcal{J}$: Submits $\mathcal{D}_{wm}$, $\mathcal{M}_{own}$ and claims ownership
 8: $\mathcal{A} \rightarrow \mathcal{J}$: Submits suspect model $\mathcal{M}_{adv}$
 9: $\mathcal{J}$: Retrieves hash $h_{stored} \leftarrow \text{Read}(\mathbb{B})$
10: **if** $H(\mathcal{D}_{wm}) \neq h_{stored}$ **then**
11:     **return Denied** (Hash Mismatch / Evidence Tampered)
12: **end if**
13: $\mathcal{J}$: Calculates threshold $t$ using DWT procedure
14: $\mathcal{J}$: Computes watermark score $s \leftarrow \text{Eval}(\mathcal{M}_{adv}, \mathcal{D}_{wm})$
15: **if** $s > t$ **then**
16:     **return Confirmed**
17: **else**
18:     **return Denied**
19: **end if**

---

