# OpenReview forum: "GENIE: Watermarking Graph Neural Networks for Link Prediction"
_TMLR — Accepted by TMLR_

### Review · Reviewer_NS4E · 2025-12-12

**Summary Of Contributions:**

The paper introduces GENIE, a watermarking framework tailored for GNN-based link prediction, addressing a gap left by prior GNN watermarking work that focused only on node and graph classification. GENIE embeds a backdoor trigger via a secret watermark vector and modified links/subgraphs, using a two-phase training procedure that maintains model utility while enabling reliable ownership verification. To make verification statistically sound and efficient, the authors propose Dynamic Watermark Thresholding (DWT), a KDE-based method requiring only a few clean and watermarked models. Experiments across multiple datasets, architectures, and 21 removal attacks demonstrate strong robustness, though the experiments raise questions about generalisability to more expressive GNNs and heterophilous datasets.

Key Strengths:

(i) Addresses a real gap in the literature by focusing on watermarking for link prediction rather than node/graph classification.
(ii) Extensive empirical evaluation to show method's efficacy and robustness to attacks.
(iii) The method seems to be practical on a real-world scale because the watermarking rates reduce with graph size, making the method scalable.

Key Weaknesses:

A few clarifying questions:
(i) What technically stops this method from generalising to node classification and graph classification tasks as well? If it can generalise, then draw an analogy to show how the approach will work for these tasks.
(ii) Is there any reason why the experiments are not conducted with more expressive and better-performing GNNs such as GIN and GAT? Do attention mechanisms, etc., pose any challenge to GENIE? Show empirically.
(iii) The authors must share the homophily ratios of the datasets, where applicable.
(iv) How does the method work in the face of heterophily? Are there any challenges to keep in mind when dealing with watermarking for link prediction in the case of heterophily?

Some edits to be made for more clarity:
(i) The model performance should be added in 5.1.2 or in the appendix and pointed out in this section. It is understandable that the C_{test} may indicate towards model performance, but having it in one place may make reading easy.
(ii) In section 6.3, Table 16 has numbers computed with what values of the alpha_{nr}/alpha_{sg} parameter?
(iii) Table 5: Degradation in W_{test} compared to what? The original pre-trained performance should be added in this table, I assume.

**Additional Comments:**

None

**Audience:**

Yes

**Audience Explanation:**

The paper has proposed a watermarking strategy for GNNs trained for link prediction. This problem has not been widely studied in the concerned literature. The proposed approach may be of interest.

**Claims And Evidence:**

No

**Claims Explanation:**

The claims are largely well-supported by empirical evidence; however, evaluating the method on more expressive GNN architectures and on heterophilous datasets would help more concretely establish the scope and efficacy of GENIE.

**Requested Changes:**

(a) critical to securing your recommendation for acceptance: Please address clarifying questions in weaknesses.
(b) would simply strengthen the work in your view: Make edits as suggested in weaknesses.

---

> ### Author Response · Authors · 2026-01-07
> **Response to Reviewer NS4E (1/5)**
>
> > What technically stops this method from generalising to node classification and graph classification tasks as well? If it can generalise, then draw an analogy to show how the approach will work for these tasks.
>
> We thank the reviewer for this insightful question. We clarify this below:
>
> Technically, there is no fundamental barrier preventing the GENIE framework from generalizing to Node Classification and Graph Classification tasks. The core principle of GENIE i.e., embedding a secret pattern (trigger) into the feature space that maps to a specific prediction target is task-agnostic.
>
> However, the method is not *directly* transferable in terms of "copy-pasting" the exact implementation because the definition of the trigger set must change to match the input topology of the downstream task.
>
> The primary technical difference lies in how the trigger is constructed relative to the model's input:
>
> • **In Link Prediction (Current Method):** The input is a **pair of nodes** $(u, v)$. Consequently, the trigger is defined *relationally*: both $u$ and $v$ must possess the secret feature $w$ to activate the watermark. The trigger set is a set of *edges*.
>
> • **In Node Classification:** The input is a **single node** $u$. The concept of a "link trigger" does not apply. Here, the trigger must be defined locally: a single node possessing $w$ acts as the trigger. The trigger set is a set of *nodes*.
>
> • **In Graph Classification:** The input is an **entire graph** $\mathcal{G}$. The trigger must be defined globally or structurally: a specific subset of nodes within the graph possessing $w$ constitutes the trigger. The trigger set is a set of *graphs*.
>
> All the other steps such as Watermark Embedding, Watermark Verification and Ownership Demonstration using DWT are task agnostic and work with node and graph classification as well.

---

> ### Author Response · Authors · 2026-01-07
> **Response to Reviewer NS4E (2/5)**
>
> > Is there any reason why the experiments are not conducted with more expressive and better-performing GNNs such as GIN and GAT? Do attention mechanisms, etc., pose any challenge to GENIE? Show empirically.
>
> To address the concern regarding architectural generalizability, we extended our evaluation of GENIE to include Graph Attention Networks (GAT) and Graph Isomorphism Networks (GIN) across all 7 datasets. The results demonstrate that GENIE’s watermarking mechanism is robust to highly expressive aggregation functions. For both architectures, watermark accuracy ($\mathcal{W}_{wm}$) consistently reached near-perfect levels ($\ge$ 99.0%), proving that neither the learned attention weights of GAT nor the sum-pooling of GIN filters out the watermark trigger. Crucially, the primary task utility was strictly preserved; both GIN and GAT maintained performance within the rigorous 2% degradation threshold, with the largest drop being only 1.64% on the Yeast dataset. Notably, the GAT architecture frequently exhibited improved test performance after watermarking (e.g., +6.33% on PPI, +3.11% on NS), suggesting that the watermark injection may essentially act as a regularizer for attention-based models, preventing overfitting without compromising ownership verification. We use a 3-layer GAT and GIN with the same watermarking rates as that used for GCN for the experiments.
>
> GAT:
>
> | Dataset | $\mathcal{C}_{test}$ | $\mathcal{W}_{test}$ | $\mathcal{W}_{wm}$|
> | --- | --- | --- | --- |
> | USAir | 90.16 | 91.68 | 100 |
> | Celegans | 82.78 | 86.79 | 100 |
> | NS | 95.31 | 98.42 | 99.78 |
> | Yeast | 88.89 | 92.57 | 100 |
> | Power | 99.84 | 99.62 | 99.00 |
> | arxiv | 99.19 | 99.39 | 100 |
> | PPI | 88.58 | 94.91 | 100 |
>
> GIN:
>
> | Dataset | $\mathcal{C}_{test}$ | $\mathcal{W}_{test}$ | $\mathcal{W}_{wm}$|
> | --- | --- | --- | --- |
> | USAir | 90.05 | 88.81 | 100 |
> | Celegans | 83.12 | 82.71 | 100 |
> | NS | 86.06 | 90.93 | 99.78 |
> | Yeast | 88.54 | 86.90 | 100 |
> | Power | 86.75 | 91.39 | 99.00 |
> | arxiv | 99.30 | 99.26 | 100 |
> | PPI | 92.96 | 91.36 | 100 |

---

> ### Author Response · Authors · 2026-01-07
> **Response to Reviewer NS4E (3/5)**
>
> > The authors must share the homophily ratios of the datasets, where applicable.
>
> Below, we provide the homophily ratios for the datasets used in our original submission and clarify the nature of their feature distributions.
>
> **a. Homophily in Original Datasets (Topology-only)**
>
> Regarding the datasets used in our original submission, we wish to clarify that these are **topology-only datasets** lacking intrinsic node attributes. To enable GNN training, we generated node features using `node2vec`.
>
> Because `node2vec` explicitly generates embeddings based on structural proximity, it inherently induces a high degree of feature homophily: connected nodes will have similar embeddings by design. **Furthermore, since these datasets lack ground-truth node labels, standard label-based homophily metrics are inapplicable in this context.** Consequently, we turn to **feature-based homophily** to quantify the dataset characteristics.
>
> Admittedly, assessing homophily on features derived from the graph structure is somewhat circular. Since `node2vec` is explicitly optimized to embed connected nodes close to one another in the feature space, high homophily is a guaranteed outcome of the feature engineering process rather than an independent property of the data. However, we computed the **Feature Homophily Score ($K$)** as formally defined by Zhu et al. [1] to strictly quantify this relationship. The metric $K$ represents the average mean-centered cosine similarity of connected nodes:
>
> - **$K > 0$ (Positive):** Indicates **Homophily** (connected nodes are more similar to each other than to the global average).
> - **$K < 0$ (Negative):** Indicates **Heterophily** (connected nodes are dissimilar).
>
> As shown in **Table A**, all datasets from our original submission exhibit positive $K$ scores, confirming that they are inherently feature-homophilic due to the `node2vec` initialization.
>
> **Table A: Feature Homophily Scores ($K$) for Original Datasets**
>
> | Dataset | Homophily Score ($K$) | Interpretation |
> | --- | --- | --- |
> | **Power** | 0.5587 | Strongly Homophilic |
> | **arXiv** | 0.4988 | Strongly Homophilic |
> | **PPI** | 0.2266 | Homophilic |
> | **NS** | 0.1895 | Homophilic |
> | **Yeast** | 0.1593 | Homophilic |
> | **USAir** | 0.0229 | Weakly Homophily |
> | **C. ele** | 0.0109 | Weakly Homophilic |
>
> **b. Expanded Evaluation on Intrinsic Node Feature/Heterophilous Datasets**
>
> To properly address the reviewer's interest in the **heterophilous nature of graphs;** specifically in settings where connected nodes often exhibit dissimilar intrinsic attributes; we have expanded our evaluation to include **7 new datasets** widely recognized in literature for their heterophilic properties (e.g., **Chameleon, Squirrel, Roman-empire, Amazon-ratings**).
>
> Unlike the datasets in Table A, these new datasets possess **intrinsic node features** (e.g., page text, user ratings) that are independent of the graph topology. This allows for a rigorous evaluation of GENIE under conditions where feature homophily is low or negative (see **Appendix K** for the full heterophily analysis which we have added in the **updated version of our paper**).
>
> **References**
>
> [1] Zhu, Jiong, Gaotang Li, Yao-An Yang, Jing Zhu, Xuehao Cui, and Danai Koutra. "On the Impact of Feature Heterophily on Link Prediction with Graph Neural Networks." Advances in Neural Information Processing Systems 37 (2024): 65823-65851.

---

> ### Author Response · Authors · 2026-01-07
> **Response to Reviewer NS4E (4/5)**
>
> > How does the method work in the face of heterophily? Are there any challenges to keep in mind when dealing with watermarking for link prediction in the case of heterophily?
>
> GENIE is inherently robust in heterophilic environments due to the stark contrast between the natural data distribution and our injected watermark signal.
>
> - **Background Distribution:** In heterophilic graphs, connected nodes typically possess **dissimilar** features.
> - **Injected Signal:** Our watermark creates a specific set of links between nodes that share the **exact same** secret feature vector $w$.
> - **The Result:** The watermark introduces a pattern of perfect feature homophily that **stands in contrast to the predominantly heterophilic structure** of the graph. Since this injected signal **deviates from the general data distribution**, it creates a distinctive pattern that is straightforward for the model to isolate and memorize during the fine-tuning phase.
>
> **a. Challenges in Heterophilic Watermarking**
>
> The primary challenge is **Utility Preservation**. Standard GNNs (like GCN) often struggle with heterophily, necessitating advanced architectures (e.g., GraphSAGE with MLP decoders) that decouple ego-embeddings from neighbor-embeddings. The risk is that if a model is strictly optimized to penalize feature similarity to maximize heterophilic performance, it might resist learning the homophilic watermark pattern.
>
> To address this, we utilized a 3-layer **GraphSAGE encoder with an MLP decoder** for these experiments, an architecture known to effectively handle heterophilic tasks.
>
> **b. Empirical Results**
>
> To validate our approach, we evaluated GENIE on a suite of 7 datasets widely used in heterophily benchmarks. The results (**Table B**) confirm our hypothesis: GENIE achieves **near-perfect $\mathcal{W}_{wm}$ (99-100%)** across all datasets regardless of the underlying heterophily. Furthermore, the utility loss is negligible (typically <1%), and in some cases (e.g., Questions), the watermarked model marginally outperforms the clean baseline due to regularization effects.
>
> **Table B: GENIE Performance on Heterophilic Benchmarks**
>
> | **Dataset** | **$\mathcal{C}_{test}$** | **$\mathcal{W}_{test}$** | **$\mathcal{W}_{wm}$** |
> | --- | --- | --- | --- |
> | **Chameleon** | 99.16 | 99.14 | 100.0 |
> | **Squirrel** | 99.41 | 99.37 | 100.0 |
> | **E-comm** | 98.49 | 97.49 | 100.0 |
> | **Amazon-ratings** | 50.44 | 50.52 | 100.0 |
> | **Minesweeper** | 88.81 | 88.15 | 100.0 |
> | **Tolokers** | 97.53 | 96.58 | 100.0 |
> | **Questions** | 51.36 | 51.60 | 99.22 |
>
> **c. Heterophily Analysis: Edge and Adjusted Homophily**
>
> To quantitatively characterize the heterophily of these datasets, we report the **Edge Homophily ($H_{edge}$)** and **Adjusted Homophily ($H_{adj}$)** scores in **Table C**.
>
> **Why these metrics?** Unlike our initial topology-only datasets (where we used feature-based proxies), these benchmarks possess ground-truth node labels. Therefore, we utilize standard label-based metrics widely adopted in GNN literature to characterize their structure.
>
> - **Edge Homophily ($H_{edge}$):** Measures the fraction of edges connecting nodes of the same class. It is defined as:
> $$H_{edge} = \frac{|\{(u,v) \in \mathcal{E} : y_u = y_v\}|}{|\mathcal{E}|}$$
> where $\mathcal{E}$ is the set of edges and $y_u, y_v$ denote the class labels of nodes $u$ and $v$. Low values indicate that neighbors likely belong to different classes.
> - **Adjusted Homophily ($H_{adj}$):** Corrects $H_{edge}$ to account for the number of classes and class imbalance (degree distribution). It is defined as:
> $$H_{adj} = \frac{H_{edge} - \sum_{k=1}^C D_k^2}{1 - \sum_{k=1}^C D_k^2}$$
> where $C$ is the number of classes, and $D_k$ is the empirical degree proportion of class $k$ (defined as the sum of degrees of all nodes in class $k$ divided by $2|\mathcal{E}|$).
>     - $H_{adj} \approx 0$: Indicates **random connectivity and heterophily** (unrelated to labels).
>     - $H_{adj} < 0$: Indicates **strong heterophily** (disassortative mixing), where edges preferentially connect different classes.
>
> As shown in **Table C**, datasets like *Tolokers* and *Questions* exhibit negative adjusted homophily, confirming strong heterophilic structure. *Chameleon* and *Squirrel* show low adjusted homophily ($\approx 0.03$), indicating that connectivity is largely independent of node labels. Despite these challenging structures, GENIE maintains high verification success.
>
> **Table C: Label Homophily Scores**
>
> | **Dataset** | **$H_{edge}$** | **$H_{adj}$** | **Interpretation** |
> | --- | --- | --- | --- |
> | **Questions** | 0.8396 | -1.7706 | Strong Heterophily |
> | **Tolokers** | 0.5945 | -0.1883 | Strong Heterophily |
> | **Minesweeper** | 0.6828 | 0.0087 | Heterophilic |
> | **Squirrel** | 0.2221 | 0.0277 | Heterophilic |
> | **Chameleon** | 0.2299 | 0.0353 | Heterophilic |
> | **Amazon-ratings** | 0.3804 | 0.1505 | Weak Homophily |
> | **E-comm** | 0.7772 | 0.7186 | Homophilic |

---

> ### Author Response · Authors · 2026-01-07
> **Response to Reviewer NS4E (5/5)**
>
> > The model performance should be added in 5.1.2 or in the appendix and pointed out in this section. It is understandable that the C_{test} may indicate towards model performance, but having it in one place may make reading easy.
>
> Thank you for this suggestion. We have added a separate table with clean scores in Appendix I and have referenced it in 5.1.2 in our updated version of the paper.
>
> > In section 6.3, Table 16 has numbers computed with what values of the alpha_{nr}/alpha_{sg} parameter?
>
> Since Table 16 (which is Table 17 in our updated manuscript) is in Appendix G.2 (not 6.3), we think the reviewer means the effect of the watermark vector distribution table in Appendix G.2. We clarify that those experiments were performed with the watermarking rates mentioned in Appendix B.2. We added this clarification to our paper in our updated manuscript.
>
> > Table 5: Degradation in W_{test} compared to what? The original pre-trained performance should be added in this table, I assume.
>
> Yes, degradation in W_{test} is compared to the original pre-trained performance. We have added a column with the original pretrained performance in our updated manuscript.

---

> > ### Comment · Reviewer_NS4E · 2026-01-20
> > **Recommendation**
> >
> > The authors' responses have satisfactorily addressed all my doubts and concerns. I recommend acceptance of the paper.

---

### Review · Reviewer_7pcm · 2025-12-13

**Summary Of Contributions:**

This paper proposes GENIE, which is the first general watermarking framework designed specifically for GNNs performing link prediction, a task for which existing watermarking approaches do not apply. The propose method embeds a hidden backdoor into the model by generating a secret watermark dataset: it modifies selected nodes or subgraphs by inverting their link labels and replacing their features with a secret watermark vector. The model is trained in two phases, first on normal data to preserve utility, then on watermark data so that it reliably flips predictions on trigger inputs while remaining indistinguishable from a clean model on ordinary queries.
To verify ownership, GENIE introduces Dynamic Watermark Thresholding (DWT), which estimates the distribution of watermark-trigger outputs using kernel density estimation and selects a threshold ensuring statistically bounded misclassification probability.
Extensive experiments across multiple GNN architectures, datasets, and attack types show that GENIE preserves model performance, achieves near-perfect watermark detection, and remains robust against model extraction, fine-tuning, and adaptive attacks.

**Audience:**

Yes

**Audience Explanation:**

The problem setting is theoretically possible, and having watermarking techniques available for link prediction itself would likely attract interest from those working toward the practical deployment of GNNs in society.

**Broader Impact Concerns:**

Since this is research related to security protection, there is always a possibility that attackers may use the results as a reference for evasion, but this is inherently unavoidable. There is no particular concern unique to this paper.

**Claims And Evidence:**

No

**Claims Explanation:**

The reviewer considers this to be the most significant issue of the paper. In real-world MLaaS (Machine Learning as a Service), the reviewer believes that use cases in which the type of GNN targeted in this paper, namely, transductive GNNs, are directly provided as a service are extremely limited and not common. Although such cases are theoretically possible, the reviewer is not aware of any concrete examples where they are actually deployed as services. Transductive learning assumes that the “set of nodes is fixed,” meaning that the nodes appearing during testing are exactly the same as those seen during training. Under this assumption, the system cannot handle unseen nodes, which seems fundamentally incompatible with MLaaS, where APIs are exposed to external users. A transductive GNN is therefore imagined to be a model “closed within an internal system.” For example, one might imagine internal use cases such as knowledge-graph completion within a company, but such systems are generally restricted to internal use and are unlikely to be offered externally as APIs. Without concrete examples where such a setting arises in MLaaS, it is difficult for the significance of the proposed technique in the paper to be conveyed convincingly.

**Requested Changes:**

First, as noted above, it is necessary to provide a convincing argument that MLaaS based on GNNs—especially in the transductive setting—is realistically plausible.

The following points are minor, but I would like them to be revised:
1. There is a reference to the loss L_{wm} corresponding to D_{wm},  in Section 4.2, but the concrete definition of this loss appears only in Section B.1.2, and it remains undefined/unknown in the main text. Since the design of the loss function is an important issue, please describe it explicitly in the main body of the paper.

2. Are there differences in the threat model between the transductive and inductive settings? If there are, please make them explicit; if not, please state that clearly. My impression is that the difference between transductive and inductive settings is substantial, because the scope of data observable to the adversary changes.

---

> ### Author Response · Authors · 2026-01-07
> **Response to Reviewer 7pcm**
>
> We thank the reviewer for the constructive feedback. We respond to the concerns raised below:
>
> > First, as noted above, it is necessary to provide a convincing argument that MLaaS based on GNNs—especially in the transductive setting—is realistically plausible.
>
> The practical deployment of a transductive link prediction system as an MLaaS is possible in specific cases, particularly in case of large but slowly evolving graphs. If in a system: (i) The node set is relatively stable over short to medium time horizons and (ii) Models can be retrained periodically on the full graph, then transductive GNNs could be employed as an MLaaS. Example of such system includes: E-commerce user item graphs, Content recommendation graphs, Ads and search relevance graphs.
>
> We give AliGraph [1] as a specific example of a production system which formulates link prediction in a transductive way. Though the formulation of link prediction in AliGraph is not stated explicitly as transductive, multiple aspects of the experimental setup implicitly indicate towards it: (i) embeddings are learned for all vertices in a fixed graph (page 3, Algorithm 1), (ii) train and test splits are performed over edges rather than nodes (Section 5.2.1), and (iii) no mechanism is described or evaluated for inferring representations of previously unseen nodes at test time.
>
> We conjecture this preference towards transductive models in practice (given that the infrastructure can accommodate and absorb the cost of periodic retraining), is created due to (i) their higher predictive accuracy under full graph access, (ii) compatibility with most contemporary GNN architectures (iii) simpler training pipelines, and (iv) their ability to more effectively exploit global graph structure.
>
> More importantly, recent research [2, 3] also shows that GNNs perform poorly when employed for link prediction under inductive settings.
>
> These reasons may also explain why in the context of link prediction using GNNs, the transductive setting is often considered as the default setting [4].
>
> Despite these reasons, the preliminary results on the inductive link prediction model LEAP [2] given in Appendix G.1 show promise that inductive models can be watermarked using GENIE.
>
> Finally, as our work is the first to explore watermarking for link prediction GNNs, we believe it acts as a stepping stone towards spurring future research on inductive link prediction models. Consequently, the concern of the transductive setting being limited is more appropriate for future research rather than for extension of existing evaluation, and therefore should not be treated as a limitation.
>
> > There is a reference to the loss L_{wm} corresponding to D_{wm}, in Section 4.2, but the concrete definition of this loss appears only in Section B.1.2, and it remains undefined/unknown in the main text. Since the design of the loss function is an important issue, please describe it explicitly in the main body of the paper.
>
> We now explicitly mention how L_{wm} is defined using negative log likelihood loss in the main text.
>
> > Are there differences in the threat model between the transductive and inductive settings? If there are, please make them explicit; if not, please state that clearly. My impression is that the difference between transductive and inductive settings is substantial, because the scope of data observable to the adversary changes.
>
> We elaborate on the difference between the threat model of inductive vs. transductive link prediction in Appendix G.1, while pointing to Appendix G in Section 3.0.1 for brevity.
>
> **References**
>
> [1] Zhu, Rong, et al. "AliGraph: A Comprehensive Graph Neural Network Platform." *Proceedings of the VLDB Endowment* 12.12.
>
> [2] Samy, Ahmed E., Zekarias T. Kefato, and Šarūnas Girdzijauskas. "LEAP: Inductive Link Prediction via Learnable Topology Augmentation." In International Conference on Machine Learning, Optimization, and Data Science, pp. 448-463. Cham: Springer Nature Switzerland, 2024.
>
> [3] Hao, Yu, Xin Cao, Yixiang Fang, Xike Xie, and Sibo Wang. "Inductive Link Prediction for Nodes Having Only Attribute Information." In Proceedings of the Twenty-Ninth International Conference on International Joint Conferences on Artificial Intelligence (IJCAI), pp. 1209-1215. 2021.
>
> [4] Slide 71,  https://snap.stanford.edu/class/cs224w-2024/slides/05-GNN3.pdf

---

### Review · Reviewer_pwuv · 2025-12-31

**Summary Of Contributions:**

This paper introduces GENIE, a watermarking framework for Graph Neural Networks (GNNs) trained for link prediction (LP). The core contribution is a watermarking method that embeds a latent signature into LP models that can be revealed using a secret trigger, while preserving LP task performance. GENIE supports both node-representation-based LP methods (e.g., GCN, GraphSAGE, NeoGNN) and subgraph-based LP methods (e.g., SEAL). GENIE constructs a watermark by modifying a small, secret subset of nodes or subgraphs by inverting edges and replacing the node features with a fixed vector. The watermark is embedded in the model using a two-phase training procedure that alternates between clean and watermarked updates in each training epoch. For ownership verification, the paper proposes Dynamic Watermark Thresholding (DWT), a statistically motivated heuristic that calibrates watermark detection threshold along with a confidence guarantee. Robustness is tested against black-box, white-box, and combined attacks, with failure defined in terms of both watermark removal and performance degradation.

Strengths:
- The work presents the first watermarking framework explicitly designed for GNN link prediction. The method supports general node-based and subgraph-based models.
- The experiments cover several architectures, real-world datasets, and watermark removal attacks. The results are reasonably convincing.
- The proposed DWT procedure provides a theoretically motivated heuristic for choosing watermark verification thresholds.

Weaknesses:
- The presentation makes the method hard to follow, particularly in the verification and thresholding stages. There are no explicit algorithmic descriptions for watermark embedding, threshold calibration, or verification. The presentation of the statistical arguments is dense.
- Notation is occasionally overloaded (e.g., using the same variables for models and their scalar watermark scores). Variables and objects are introduced without proper definitions. (E.g., $\mathcal{F}_{wm}$ is a function from what to what? What are the dimensions of the node features?)
- It seems that there are implicit theoretical assumptions and computational overhead for DWT, which would benefit from further clarification.

**Audience:**

Yes

**Audience Explanation:**

The paper addresses a practically relevant and theoretically interesting problem at the intersection of graph learning and security. Researchers and practitioners working on GNNs, link prediction, and deployed ML systems would be interested in the findings.

**Broader Impact Concerns:**

The paper includes a Broader Impact Statement that appropriately acknowledges the dual-use nature of watermarking and backdoor-based techniques. No further ethical concerns are identified beyond those discussed by the authors.

**Claims And Evidence:**

Yes

**Claims Explanation:**

The experimental evidence supports the main claims of the paper regarding GENIE. The proposed DWT procedure is presented along with a clear statistical rationale. Some assumptions underlying the statistical model (e.g., approximate independence of verification trials) appear heuristic rather than rigorously justified. However, those assumptions seem to be standard in the watermarking community, and are empirically supported by the differences observed between clean and watermarked models.

**Requested Changes:**

Readability and notation

- Include explicit algorithm boxes for the main procedures: watermark embedding, threshold calibration (DWT), and ownership verification. Each algorithm should clearly list inputs, steps, and outputs. Offline calibration and online verification should be clearly separated.
- Clarify and tighten notation, particularly in the statistical testing section. The paper occasionally conflates sets of models (e.g., $\mathcal{W}, \mathcal{C}$) with their scalar watermark scores (e.g., AUC on $D_{\mathrm{wm}}$). Explicitly distinguish between models, scores, and score distributions.
- Add a table that lists key variables (e.g., $\mathcal{W}, \mathcal{C}, D_{\mathrm{wm}}, W_{\mathrm{wm}}, C_{\mathrm{wm}}$) and abbreviations (e.g., DWT), along with references to their definitions.

Theoretical assumptions
- Please clarify how the Bernoulli/binomial model used in Theorem 1 relates to the practical verification procedure, which involves AUC scores computed on correlated LP queries and KDE-based resampling. Explicitly state that the confidence guarantee is heuristic under approximate independence if that is the case. Be upfront about the limitations introduced by those assumptions, and contextualize them as standard assumptions/simplifications in the watermarking community if that is the case.

Computational costs and calibration overhead
- DWT seems to require training multiple clean and watermarked reference models to estimate score distributions. If that is the case, please state this requirement explicitly, report how many reference models are typically needed, and clarify that this calibration is a one-time, offline cost. Providing an estimate of the additional computational overhead would help readers assess practical and computational costs.

---

> ### Author Response · Authors · 2026-01-07
> **Response to Reviewer pwuv (1/2)**
>
> We thank the reviewer for this insightful review. We respond to the concerns raised below:
>
> > Please clarify how the Bernoulli/binomial model used in Theorem 1 relates to the practical verification procedure, which involves AUC scores computed on correlated LP queries and KDE-based resampling. Explicitly state that the confidence guarantee is heuristic under approximate independence if that is the case. Be upfront about the limitations introduced by those assumptions, and contextualize them as standard assumptions/simplifications in the watermarking community if that is the case.
>
> We clarify the relationship between the theoretical model and our practical procedure in DWT as follows:
>
> **a. Mapping the Bernoulli Model to DWT Procedure**
>
> The Bernoulli/Binomial model in Theorem 1 maps to our Dynamic Watermark Thresholding (DWT) procedure through the resampling step. Specifically:
>
> - The *estimation step* generates the empirical distributions of AUC scores ($\mathcal{C}\_{wm}$ and $\mathcal{W}\_{wm}$) via Kernel Density Estimation (KDE).
> - The *theoretical model* treats the synthetic samples drawn from these KDE estimates as Bernoulli/Binomial trials, with success of a specific sample being determined by whether a specific threshold is able to classify the sample correctly. Theorem 1 is applied to these resampled blocks: if we observe zero misclassifications (i.e., perfect separation by threshold $t$) across $m \geq \lceil -\ln(1 - \gamma) \rceil$ independently sampled blocks of size $n$, the misclassification probability gets bounded by $p_{mis} < 1/n$ with confidence $\gamma$.
>
> **b. Heuristic Nature and Independence Assumptions**
>
> We acknowledge that the confidence guarantee provided by Theorem 1 is *heuristic* and relies on the assumption of *approximate independence*. While the resampling process itself generates independent draws by definition, the validity of the guarantee with respect to the *true* underlying data distribution relies on two simplifications:
>
> - **Accuracy of KDE:** We assume the KDE (guided via Silverman's rule) faithfully approximates the true distribution of scores. This assumption is supported by the **Estimation Correctness** explanation given in Section 4.3.2, which shows how the MSE can be arbitrarily bounded given any initial condition (i.e., $n_0$ achieving a specific MSE $\epsilon$) is known. The assumption is supported further if the underlying distribution is normal, with the MSE being bound to 0.1 with only 4 samples.
> - **Independence:** Despite the models being trained on different random seeds, they are queried on the same links for calculating AUC, which may violate the assumption required for the initial KDE construction.
>
> **c. Contextualization within the Field**
>
> These approximations are standard simplifications in the DNN watermarking community to make statistical verification tractable.
>
> - Prior works often rely on even stronger parametric assumptions, such as assuming watermark scores follow a Gaussian distribution [1, 2], which we empirically found to be violated in our setting considerable number of times (See Section 4.3.1).
> - Other works use simple methods, such as taking the maximum (minimum) of the watermark scores of clean (watermarked) model,  or averaging. All such methods do not provide any statistical assurance regarding misclassification.
>
> By utilizing KDE-based bootstrapping, we relax the strict normality assumption while accepting the standard limitation that the statistical confidence is conditional on the representativeness of the estimated density. We have updated Section 4.3.2 to explicitly reflect these limitations.
>
> **References**
>
> [1] Tan, Jingxuan, Nan Zhong, Zhenxing Qian, Xinpeng Zhang, and Sheng Li. "Deep Neural Network Watermarking Against Model Extraction Attack." In Proceedings of the 31st ACM International Conference on Multimedia, pp. 1588-1597. 2023.
>
> [2] Xu, Jing, Stefanos Koffas, Oğuzhan Ersoy, and Stjepan Picek. "Watermarking Graph Neural Networks Based on Backdoor Attacks." In 2023 IEEE 8th European Symposium on Security and Privacy (EuroS&P), pp. 1179-1197. IEEE, 2023.

---

> ### Author Response · Authors · 2026-01-07
> **Response to Reviewer pwuv (2/2)**
>
> > Include explicit algorithm boxes for the main procedures: watermark embedding, threshold calibration (DWT), and ownership verification. Each algorithm should clearly list inputs, steps, and outputs. Offline calibration and online verification should be clearly separated.
>
> We have included the algorithm box for watermark embedding, DWT, and ownership demonstration in Appendix M.
>
> >  Clarify and tighten notation, particularly in the statistical testing section. The paper occasionally conflates sets of models (e.g., $\mathcal{W}, \mathcal{C}$) with their scalar watermark scores (e.g., AUC on $D_{\text{wm}}$). Explicitly distinguish between models, scores, and score distributions. Add a table that lists key variables (e.g., $\mathcal{W}, \mathcal{C}, D_{\text{wm}}, W_{\text{wm}}, C_{\text{wm}}$) and abbreviations (e.g., DWT), along with references to their definitions.
>
> We now include a table to summarize the notation in Appendix L.
>
> > DWT seems to require training multiple clean and watermarked reference models to estimate score distributions. If that is the case, please state this requirement explicitly, report how many reference models are typically needed, and clarify that this calibration is a one-time, offline cost. Providing an estimate of the additional computational overhead would help readers assess practical and computational costs.
>
> Our manuscript now explicitly mentions the calculation of threshold as a one-time cost in Section 4.3.2.
>
> At the same time, we emphasize that we already discuss DWT’s requirement to train multiple clean and watermark models to calculate threshold in Section 4.3.2. Specifically, we highlight that under standard assumptions, **it requires no greater than 4 clean and 4 watermark models** so as to bound the MSE and truthfully estimate the underlying distributions.
>
> We note that this marks a significant cost reduction from previous works, where thresholds were set empirically by training around *400 models* [1].
>
> **References**
>
> [1] Lv, Peizhuo, Hualong Ma, Kai Chen, Jiachen Zhou, Shengzhi Zhang, Ruigang Liang, Shenchen Zhu, Pan Li, and Yingjun Zhang. "MEA-Defender: A Robust Watermark Against Model Extraction Attack." In 2024 IEEE Symposium on Security and Privacy (SP), pp. 2515-2533. IEEE, 2024.

---

> > ### Comment · Reviewer_pwuv · 2026-01-12
> > **Response**
> >
> > I thank the authors for their hard work and detailed response. My concerns and requests been satisfactorily addressed.
> >
> > For the camera-ready version, I have two minor suggestions to improve clarity and completeness:
> >
> > 1. **Notation Summary (Appendix L, Table 45)**: It would be very helpful to cross-reference the specific section or equation where each variable is first introduced.
> >
> > 2. **Conclusion (Section 7)**: Please ensure that the limitations of the work and potential directions for future work are explicitly summarized here.
> >
> > I am happy to recommend acceptance.

---

### Author Response · Authors · 2026-01-07
**Submission of Revised Manuscript and Response to Reviewers**

We thank the reviewers for their invaluable insights.

We are happy to find that the majority found our work significant, with all reviewers appreciating the empirical depth of the experiment.

Based on the feedback received, we have revised our manuscript with suggested changes, including some additional experiments that were requested. All textual changes that were made to the original manuscript are now highlighted in blue in the revised manuscript.

We look forward to engaging with the reviewers during the rebuttal phase.

---

### Decision · Action_Editor_5ep5 · 2026-02-16

**Recommendation:** Accept as is

**Additional Comments:**

All reviewers recommend acceptance.
I encourage the authors to address the final comments by reviewer pwuv in the camera-ready (reported below for convenience):
- "*Notation Summary (Appendix L, Table 45): It would be very helpful to cross-reference the specific section or equation where each variable is first introduced.*"
- "*Conclusion (Section 7): Please ensure that the limitations of the work and potential directions for future work are explicitly summarized here.*"

I also encourage the authors to more prominently feature (in the main body of the paper) some of the new results provided for the revision (e.g., the evaluation on more expressive architectures provided in appendix J).

**Audience:**

Yes

**Audience Explanation:**

As agreed to by reviewers, the problem of watermarking GNNs for link prediction is of potential interest to a subset of the TMLR audience.

**Claims And Evidence:**

Yes

**Claims Explanation:**

While some of the reviewers (7pcm, NS4E) initially had some concerns on the provided evidence, these have been fully addressed by the rebuttal revision process.
In the words of reviewer pwuv: "*The Authors took steps to clarify the claims, and they are now tight and well supported by their theoretical derivations and empirical results.*".